# A corrosion inhibiting layer to tackle the irreversible lithium loss in lithium metal batteries

Chengbin Jin [1] ✉, Yiyu Huang[1], Lanhang Li[1], Guoying Wei[1], Hongyan Li[1], Qiyao Shang[2], Zhijin Ju[2], Gongxun Lu[2], Jiale Zheng[2], Ouwei Sheng[3] ✉ & Xinyong Tao [2] ✉

Reactive negative electrodes like lithium (Li) suffer serious chemical and electrochemical corrosion by electrolytes during battery storage and operation, resulting in rapidly deteriorated cyclability and short lifespans of batteries. Li corrosion supposedly relates to the features of solid-electrolyte-interphase (SEI). Herein, we quantitatively monitor the Li corrosion and SEI progression (e.g., dissolution, reformation) in typical electrolytes through devised electrochemical tools and cryo-electron microscopy. The continuous Li corrosion is validated to be positively correlated with SEI dissolution. More importantly, an anti-corrosion and interface-stabilizing artificial passivation layer comprising low-solubility polymer and metal fluoride is designed. Prolonged operations of Li symmetric cells and Li||LiFePO$_4$ cells with reduced Li corrosion by ~74% are achieved (0.66 versus 2.5 μAh h$^{-1}$). The success can further be extended to ampere-hour-scale pouch cells. This work uncovers the SEI dissolution and its correlation with Li corrosion, enabling the durable operation of Li metal batteries by reducing the Li loss.

With the approaching of an electrified era, rechargeable batteries have been widely adopted, which greatly change the patterns of daily life and industries[1]. To satisfy the multiple concerns of energy, environment, and resource[2], expeditions on novel and powerful battery systems boom[3–7]. Lithium (Li) metal battery emerges as the next-generation energy storage technique, which holds promise to break through the specific energy limit (>500 Wh kg$^{-1}$) of conventional battery systems[8,9]. Substantial progress has been made in the development of Li metal batteries by tackling the typical issues of highly reactive Li metal, mainly including Li dendrite growth[10], accumulation of inactive Li[11], and unstable electrode/electrolyte interfaces[12]. However, the stability and lifespan of Li metal batteries still remain formidable challenges, which are highly related to the irreversible Li loss associated with incomplete Li stripping, interface evolutions, and corrosion behaviors.

Corrosion is an irreversible system phenomenon of materials (e.g., structural metal, ceramic, and polymer)[13–15], involving such diverse factors as material, coating, environment, microbiology, stresses, and/or electromagnetism (Supplementary Notes 1 and 2, and Supplementary Table 1). Typically, the corrosion of a metal refers to the oxidation of the metal to its ionic species and release of electrons (e.g., rusting of steel), which challenges the longevity, safety, and function of products[16]. Li as a reactive metal is highly susceptible to corrosion, which typically is independent of extra external current or potential polarization, and contributes to the self-discharging of batteries (Supplementary Note 3)[17,18]. The terminology of corrosion in battery research dates back to 1979 when Peled et al. described the solid-electrolyte-interphase (SEI, i.e., a layer of corrosion product) at the Li metal–liquid electrolyte interface[19]. Similar to the corrosion of structural metals[20,21], Li corrosion highly relates to the features (i.e.,

[1]College of Materials and Chemistry, China Jiliang University, Hangzhou 310018, China. [2]College of Materials Science and Engineering, Zhejiang University of Technology, Hangzhou 310014, China. [3]Institute of Advanced Magnetic Materials, College of Materials and Environmental Engineering, Hangzhou Dianzi University, Hangzhou 310012, China. ✉e-mail: jincb@cjlu.edu.cn; owsheng@hdu.edu.cn; tao@zjut.edu.cn

composition, structure, morphology, and basic functions) and progression (e.g., dissolution and reformation) of such a passivation layer (i.e., SEI). Very recently, Cui and Zhang et al. have validated the SEI dissolution in alkali metal battery systems[22,23]. In detail, the native SEI suffers a dynamic progression of repeated swelling, dissolution, breakage, and reformation, leaving Li with high redox power exposed to electrolytes. Consequently, the direct chemical depletion of both electrolyte and active Li sources is constantly undergone. Moreover, Li corrosion can be electrochemically aggravated by forming localized galvanic couples comprising dissimilar metals (e.g., active Li and noble copper) or regions (e.g., corroded pit and passivated regions nearby) in electrical contact and electrolyte[24,25]. The chemical and electrochemical corrosion patterns continuously undergo during battery storage and operation, together contributing to serious Li loss[26]. The Li corrosion and SEI dissolution emerge as additional failure mechanisms of reactive Li metal negative electrodes.

Numerous expeditions have been devoted to circumventing corrosion mainly by engineering SEI, including formulating electrolytes (e.g., fluorinated solvents and additives) and introducing artificial SEI (e.g., films of nitrides, carbon materials, self-assembled monolayers)[27–32]. Though some progress has been made to stabilize SEI and conveniently inhibit Li corrosion, the improvement of battery lifespan is still far from ideal. Such a dilemma highly relates to the limited recognition of the Li corrosion science, and corrosion-induced Li loss cannot be eliminated[33–35], appealing for more expeditions to uncover the correlation between dynamic Li corrosion and SEI progression (Supplementary Table 2). The corrosion science in battery systems needs to be understood and established (Supplementary Notes 4 and 5).

In this work, Li corrosion behaviors referring to both chemical and electrochemical corrosion have been quantitatively monitored and revealed by devised electrochemical tools along with cryo-electron microscopy. The underlying correlation among Li corrosion, SEI dissolution, and battery failure has been understood. The dissolution of native SEI is proved to be responsible for the continuous Li corrosion. From a corrosion science perspective, a shielding strategy by employing a functional passivation layer of low-solubility polymer and embedded metal fluoride is designed to reduce Li corrosion by 74%. Consequently, Li‖Cu cells with superiorly stable interfaces are enabled to deliver high average Coulombic efficiency (CE, 96.2% for 500 cycles at 1.0 mA cm$^{-2}$), and the optimized Li‖LiFePO$_4$ cell exhibits superior cycling stability (1500 cycles at 1.3 mA cm$^{-2}$). Such a shielding strategy can be preliminarily translated into real ampere-hour-scale Li‖LiFePO$_4$ and Li‖LiNi$_{0.5}$Co$_{0.2}$Mn$_{0.3}$O$_2$ (NCM523) pouch cells. The role of artificial SEI has been extendedly understood from the perspective of anti-corrosion, which distinguishes it from conventional designs. Most importantly, this work uncovers the dynamic Li corrosion and SEI dissolution, which promises Li metal batteries with improved lifespans by reducing the corrosion-induced Li loss.

## Results and discussion
### Li corrosion behavior and its correlation with SEI dissolution
SEI as the passivation layer of Li corrosion by electrolyte cannot totally eliminate the side reactions at the Li metal−liquid electrolyte interface. Consequently, corroded Li deposits on the current collector with damaged morphologies after short durations of rest in the electrolytes can be frequently monitored (Fig. 1a–d)[34]. Such Li deposits suffering chemical and/or electrochemical corrosion (e.g., pitting corrosion and galvanic corrosion) can be found in typical ether and ester electrolytes (Fig. 1e and Supplementary Figs. 1 and 2). Obviously, Li corrosion will bring in the loss of active Li and reduced CE, which can be quantitively determined by a facile electrochemical protocol. In detail, galvanostatic Li plating was conducted in a typical Li‖Cu cell, while after a duration of rest (0, 5, 15, 50, 300, and 500 h), Li was totally stripped (Supplementary Fig. 3a). Notably, the capacity loss of Li deposits

without rest indicates the consumption during initial SEI formation (~66 μAh) (Fig. 1f). Subsequently, the continuous chemical and electrochemical Li corrosion proceeds originating from the inferiority and progression of initially formed SEI, which may contribute to the accumulation of dead Li and increase of cell impedance. Notably, cells with a resting time within 50 h delivered rapidly enlarged capacity loss of Li deposits and deteriorated CE (125 μAh, and 89% for 50 h). With the increasing resting time, Li corrosion continues to induce the increase of capacity loss and deterioration of CE (170 μAh, and 85% for 500 h), while the changes become much slower. Since Li corrosion highly relates to SEI progression, the latter of which was further studied by another electrochemical protocol. The repeated galvanostatic charging/discharging (2−0.05 V versus Li/Li$^+$) and extended open-circuit pauses were conducted in Li‖Cu cell[36]. Note that no Li deposits formed during this test above 0 V vs. Li/Li$^+$. The cell was firstly cycled to form a relatively stable SEI at the initial cycles, after which variable durations of rest were conducted (Supplementary Fig. 3b). Extra capacity is found to be required after rest to supplement the SEI loss probably caused by dissolution, though such capacity loss is much smaller than that of direct Li corrosion. Notably, the SEI dissolution can be detected in both ether and ester electrolytes (Supplementary Fig. 3c)[22,37]. Moreover, the SEI dissolution was directly validated by detecting the morphological, structural, and chemical changes of SEI before and after soaking in the electrolyte. The SEI suffered obvious shrinkage after soaking in the electrolyte (Fig. 1g, h, and Supplementary Figs. 4–6). Additionally, the SEI after soaking contained abundant inorganic Li salts and limited polymeric composition, indicating the higher solubility of organic components (Fig. 1i and Supplementary Fig. 7). To confirm such hypothesis, the dissolved SEI compositions in the DOL/DME were collected and studied, which were found to be comprised of a large number of amorphous polymeric substances and a few crystalline Li salts (Fig. 1j). A reasonable description for the Li corrosion mechanism is that fresh Li will be repeatedly exposed to the electrolyte due to the SEI dissolution, triggering continuous side reactions. Such progress further induces the net growth of flocculent SEI and seriously deteriorates the lifespans of Li metal batteries (Supplementary Figs. 3d and 8)[34]. A Li metal−liquid electrolyte interface with superior features (e.g., low solubility and high ionic conductivity) during battery storage and operation should be established to prevent Li corrosion.

### Design and fabrication of an artificial passivation layer against Li corrosion
A primary idea for tackling the SEI dissolution is to introduce a low-solubility "shield" (i.e., an artificial passivation layer), which can effectively withstand the swelling and solvation of electrolytes to eliminate Li corrosion. Besides, this passivation layer should exhibit superior mechanical and (electro)chemical stability and facilitate fast ion transport. Such concerns promote the application of an artificial passivation layer comprised of low-solubility poly(vinylidene fluoride) (PVDF) and metal fluoride (MF$_x$@PVDF, M = Li, Na, Mg, Al, Zn, La, and Ca, $x$ = 1, 2, or 3)[38,39]. The PVDF matrix is supposed to inhibit the Li corrosion by isolating electrolytes and electron tunneling. The embedded metal fluorides promise the formation of LiF and lithiophilic alloy (e.g., Li−Mg) at the interface, which promotes superior ion transport and uniform Li deposition[40]. By a facile coating methodology (Fig. 2a), the polymer solution in N-methyl-2-pyrrolidinone (NMP) with/without fluorides can be attached to the commercial Li foil (MF$_x$@PVDF-Li) (Supplementary Fig. 9) or Cu foil/foam (MF$_x$@PVDF-Cu) (Supplementary Figs. 10–12). MgF$_2$ as a typical low-solubility and lithiophilic fluoride, was selected as a prototype during the investigation of the artificial passivation layer (Supplementary Fig. 13)[41]. Note that the optimized content of MgF$_2$ was decided to be 30 wt% (Supplementary Fig. 14). As shown in the scanning electron microscopy (SEM) images, a uniform layer with a thickness of ~25 μm was tightly

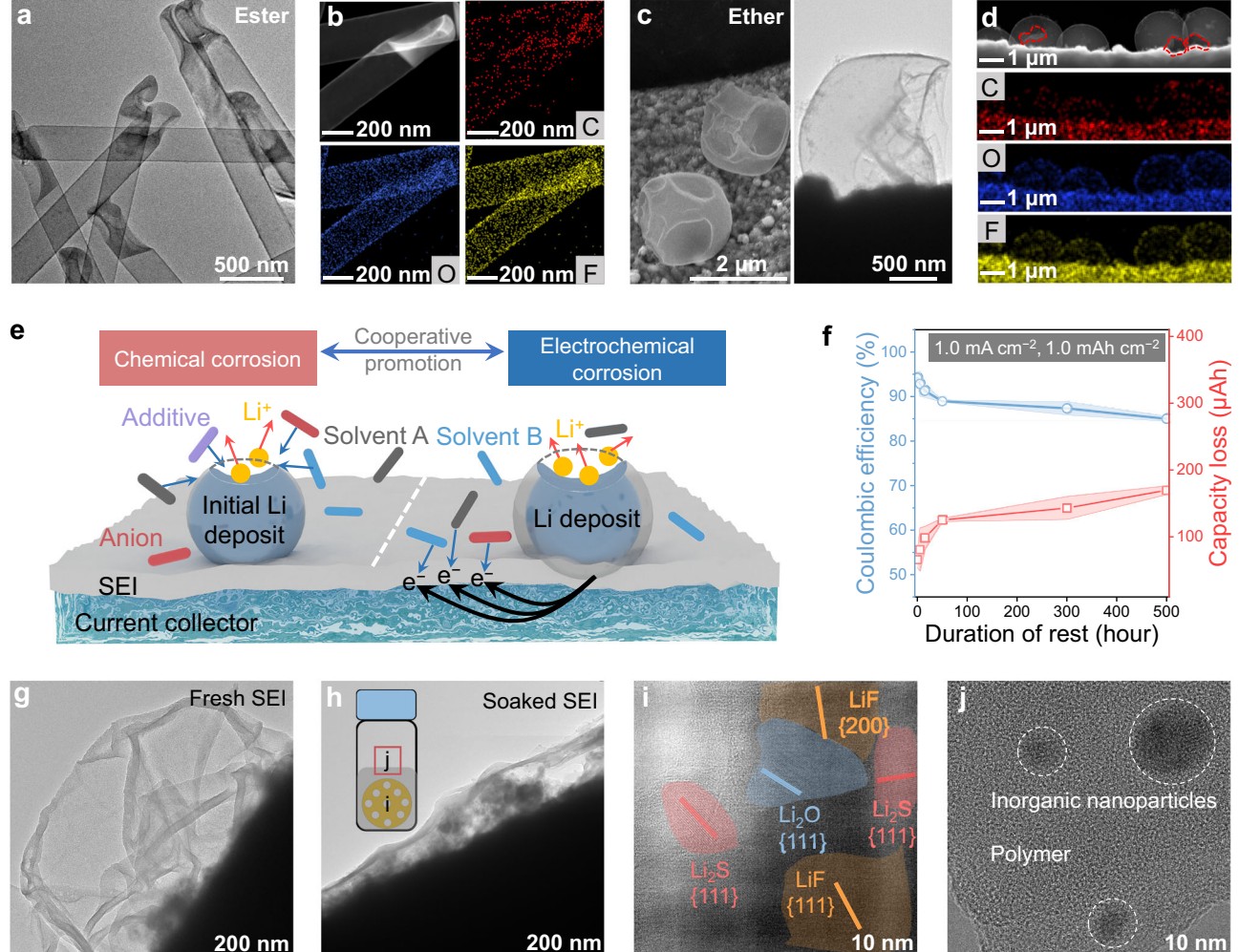

**Fig. 1 | Li corrosion and SEI dissolution. a–d** Cryo-TEM images and elemental mappings of Li deposits in 1.0 M lithium hexafluorophosphate (LiPF₆) in ethylene carbonate (EC)/ethyl methyl carbonate (EMC)/diethyl carbonate (DEC) (v/v/v = 1:1:1) with 1.0 wt% fluoroethylene carbonate (FEC), and 1.0 M lithium bis(trifluoromethanesulphonyl)imide (LiTFSI) in 1, 3-dioxolane (DOL)/1, 2-dimethoxyethane (DME) (v/v = 1:1) with 1.0 wt% LiNO₃, respectively. **e** A schematic illustration of the chemical and electrochemical corrosion of Li metal. **f** Corrosion-related capacity loss of Li and the corresponding CE as a function of rest time. The error band corresponded to the standard deviation from the measurements with three identical cells. **g, h** Cryo-TME images of SEI before and after soaking in the electrolyte. **i** HRTEM of SEI after soaking in the electrolyte. The electrolyte was 1.0 M LiTFSI in DOL/DME with 1.0 wt% LiNO₃. **j** HRTEM images of precipitation after drying the DOL/DME solvent containing dissolved SEI.

produced on the surface of Li in contrast to a flat surface of pristine Li foil (Fig. 2b, c, and Supplementary Fig. 15). Additionally, energy dispersive X-ray spectroscopy (EDS) indicated the excellent compatibility between Li metal and the artificial passivation layer (Fig. 2d, e). Such an artificial passivation layer holds promise to uptake the electrolyte and eliminate Li corrosion for durable operation. To determine its influence on the crystal structure and surface chemistry of Li, X-ray diffraction (XRD) and Fourier transform infrared spectroscopy (FTIR) were conducted on Li metal before and after coating. The characteristic peaks of MgF₂ (PDF#41-1443) and Li (PDF#15-0401) can be found after the introduction of an artificial passivation layer onto Li metal (Fig. 2f and Supplementary Fig. 16). In FTIR, the characteristic peaks at ~762 and 839 cm⁻¹ represented the existence of α and β phase of PVDF (Fig. 2g), respectively[42]. Elemental information on the passivated Li metal surface was further detected by X-ray photoelectron spectroscopy (XPS) (Supplementary Fig. 17). Signals of C *1s*, O *1s*, F *1s*, and Mg *2p* were fitted and presented. A small peak ascribing to LiF can be found in the F *1s* spectrum[43], originating from the reaction between Li and MgF₂. Conclusively, an artificial passivation layer of low-solubility polymer and metal fluoride is designed onto the surface of the Li metal

negative electrode or current collector, which shows the potential to interdict Li corrosion.

## The electrochemical function of artificial passivation layers on negative electrode

The direct correlation between the artificial passivation layer and the inhibited Li corrosion was first validated by the aforementioned electrochemical protocol (Fig. 3a). After a duration of rest (50 h), the capacity loss of Li with the application of MgF₂@PVDF layer (33 μAh) was quantified to be decreased by 74%. With the increase in resting time, the capacity loss of Li deposits shows a gradual increase (84 μAh for 500 h). Notably, the CE of cells with MgF₂@PVDF layers remains above 92% as a function of time, also indicating the apparently suppressed Li corrosion. In addition, the passivation of the bare PVDF layer is weaker than that of MgF₂@PVDF, delivering higher capacity loss and lower CE, which validates the synergistic effect of PVDF and MgF₂ in Li protection. An ideal artificial passivation layer should not only passivate the negative electrode but also promise sufficient interfacial kinetics. Consequently, the ionic conductivity of swollen MgF₂@PVDF and bare PVDF layer was measured. With the adoption of fluoride, the

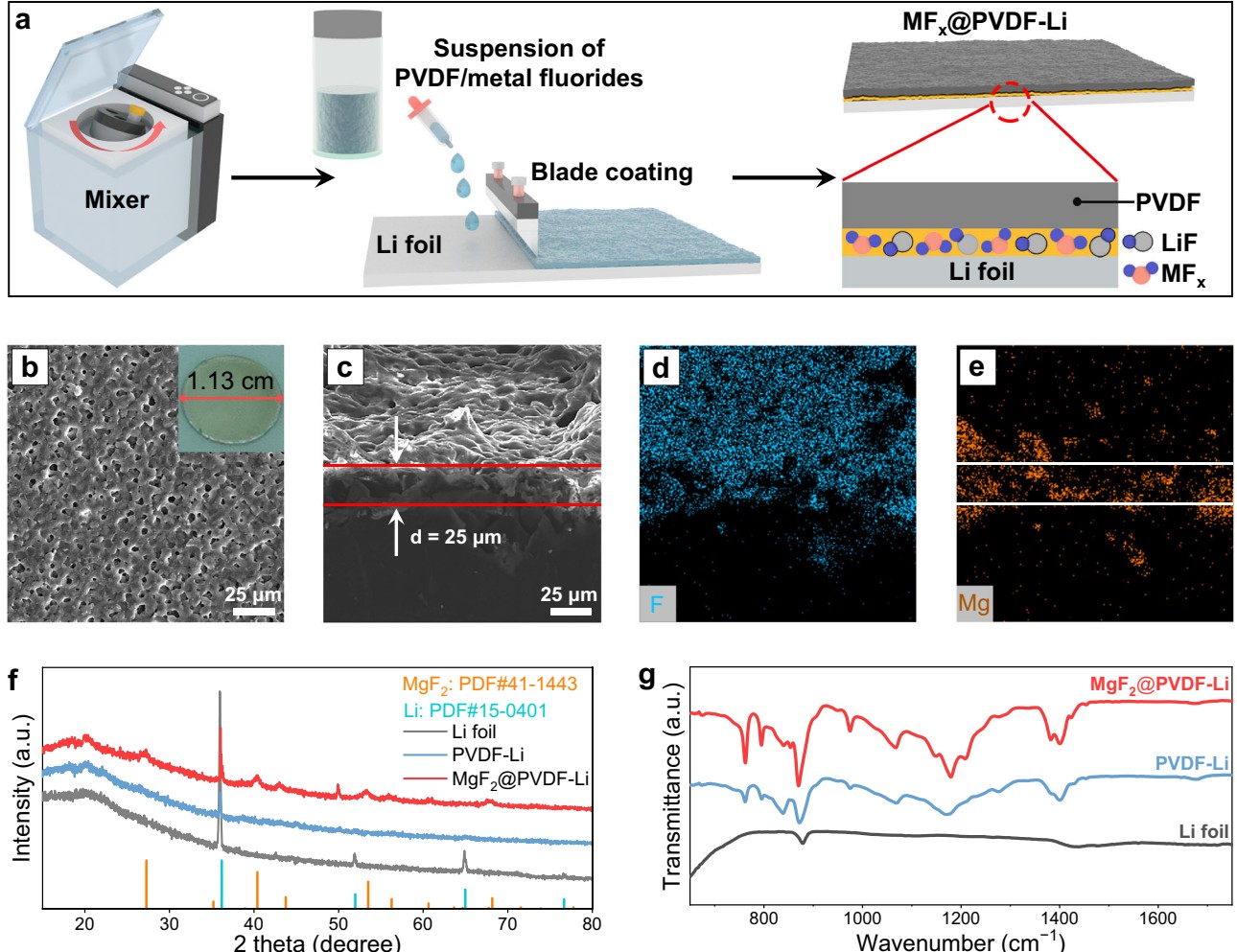

**Fig. 2 | Fabrication and characterization of MF$_x$@PVDF layer. a** Schematic illustration showing the fabrication of MF$_x$@PVDF on Li surface. **b** Top-view SEM image of MgF$_2$@PVDF-Li. **c–e** Cross-section SEM image of MgF$_2$@PVDF-Li and corresponding elemental mappings for F and Mg. **f** XRD patterns of Li foil, PVDF-Li, and MgF$_2$@PVDF-Li. **g** FTIR results of Li foil, PVDF-Li, and MgF$_2$@PVDF-Li.

ionic conductivity of PVDF layer is largely improved (~6 × 10$^{-4}$ S cm$^{-1}$ at 25 °C, Supplementary Fig. 18a), accounting for the slightly increased cell impedance (Fig. 3b). The exchange current densities as a function of cycle number were obtained from Tafel plots (Fig. 3c and Supplementary Fig. 18b), which were positively correlated with the Li$^+$ diffusion at the interface[44]. Obviously, the MgF$_2$@PVDF layer with a larger exchange current density (0.1 − 0.3 mA cm$^{-2}$) enabled the fast diffusion of Li$^+$ at the interface, benefiting the homogenous Li deposition. The gradual increase of exchange current density may be ascribed to the increase in the intrinsic rate of Li$^+$ exchange of the SEI or the electrochemically active area for Li/Li$^+$ redox[39,45].

Next, cryo-transmission electron microscopy (cryo-TEM) was employed to uncover the role of the artificial passivation layers in Li protection from perspectives of microscopic morphology and structure. The MgF$_2$@PVDF layer was directly coated onto the Cu grid for TEM, where Li was deposited (Supplementary Fig. 19). The uniform spherical Li deposits with a coating layer can be seen on the Cu grid, which showed no appearance of corrosion after resting (Fig. 3d, e). High-resolution TEM images clearly exhibited the double-layered SEI on Li deposits (Fig. 3f, g). The amorphous polymeric outer layer containing PVDF inhibited Li corrosion, while the inner inorganic layer comprised of Mg- and Li-containing species (e.g., Mg, LiMg, Li$_2$O, and LiOH) promoted rapid ion transport at the interface. In contrast, the SEI of Li deposits without the artificial passivation layer was thick and

comprised of abundant crystals (e.g., Li$_2$O) due to the Li corrosion and net growth of SEI (Supplementary Fig. 20). Electron energy loss spectroscopy (EELS) mapping was further carried out to witness the elemental distributions of multi-layered interface of Li deposits (Fig. 3h), indicating the tight adhesion of MgF$_2$@PVDF layer on the surface of Li deposits. Particularly, the polymer layer isolates Li deposits from electrolytes, promising the reduction of Li corrosion. In terms of the mixed ionic and electronic conducting layer of Mg- and Li-containing species, it promotes fast ion transport and uniform Li deposition. Such features hold promise to improve the cycling stability and extend the lifespans of cells.

### Electrochemical evaluations in Li||Cu and Li||Li cells

The function of artificial passivation layers in Li protection was primarily evaluated in Li||Cu and Li||Li cells. CE as a significant cell parameter was measured. Under moderate conditions (1.0 mA cm$^{-2}$, 1.0 mAh cm$^{-2}$), MF$_x$@PVDF-Cu delivered much higher CE and better cyclic stability than that of bare Cu electrode (CE of 85% at the 194th cycle) (Fig. 4a and Supplementary Fig. 21). Of note, PVDF-Cu had an improved CE of 94.2% at the 297th cycle. MgF$_2$@PVDF-Cu electrode delivered the highest average CE of 96.2% for over 500 cycles, which is comparable to the results in the literature on artificial SEI or protective layers (Supplementary Table 3). Similar conclusions can also be identified at larger current densities (3.0 and 5.0 mA cm$^{-2}$, Supplementary

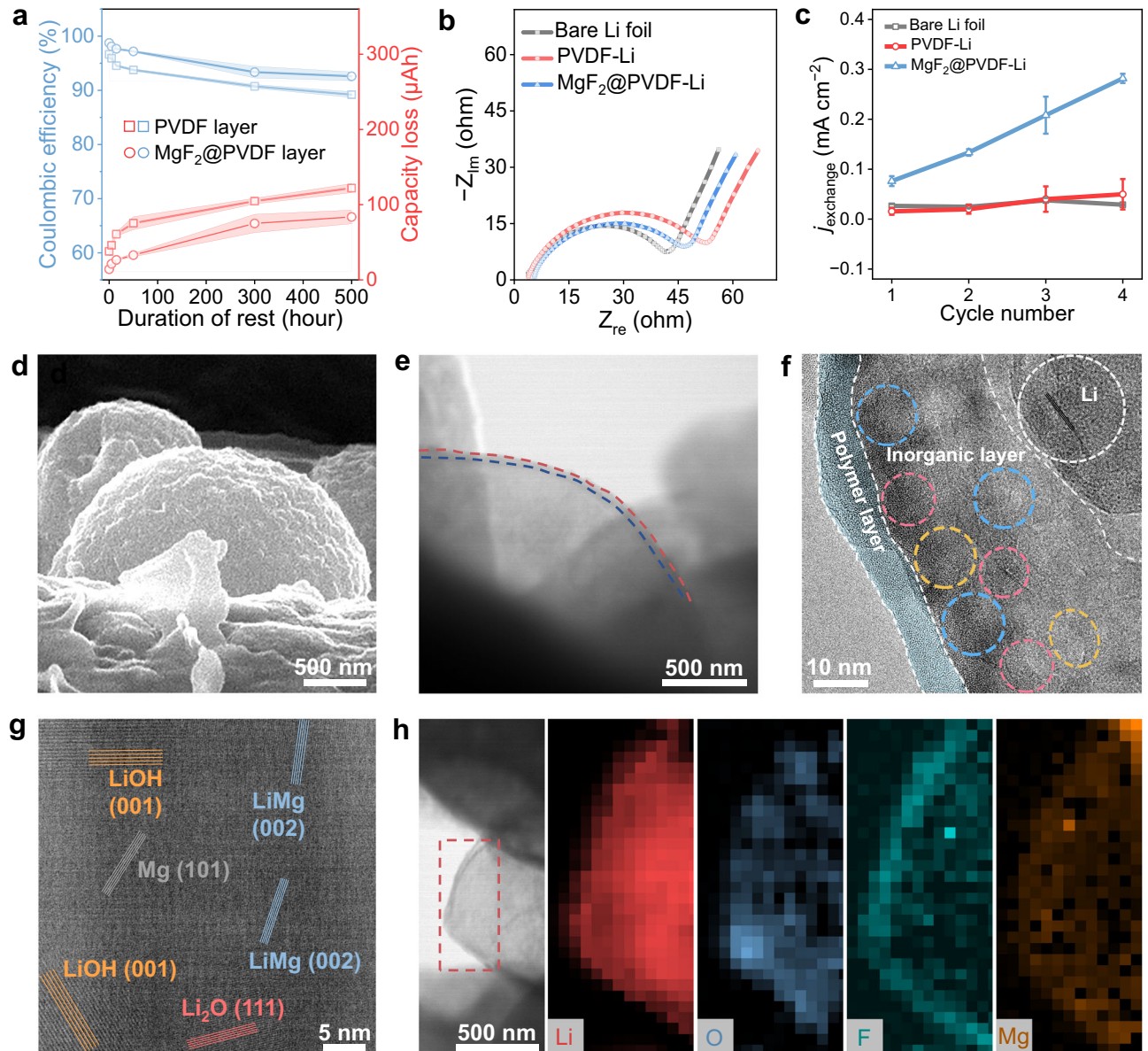

**Fig. 3 | The multiple functions of artificial passivation layers in protecting Li metal negative electrode. a** Corrosion-related capacity loss of Li and the corresponding CE as a function of rest time with the adoption of artificial passivation layers. The error band corresponded to the standard deviation from the measurements with three identical cells. **b** Impedance of fresh Li‖Cu cells with bare Li, PVDF-Li, and MgF₂@PVDF-Li. **c** Exchange current densities of cells with different Li metal negative electrodes as a function of cycle number. The error bar corresponded to the standard deviation from the measurements with three identical cells. **d** SEM image of Li deposits on Cu grid with a MgF₂@PVDF layer. **e** Cryo-TEM images of Li deposits on MgF₂@PVDF-Cu grid. **f, g** High-resolution TEM images of Li deposit on MgF₂@PVDF-Cu grid. **h** EELS maps of Li deposits on MgF₂@PVDF-Cu grid.

Figs. 22 and 23). Such improvement of CE potentially relates to the uniform Li deposition and reduced corrosion guided by Li-Mg alloy and LiF at the interface[46–49]. As shown in Fig. 4c, messy Li dendrites were observed on bare Cu foil. In sharp contrast, uniform Li deposits were plated on MgF₂@PVDF-Cu (Fig. 4d, and Supplementary Figs. 24–26). Moreover, with the adoption of MgF₂@PVDF, the Li deposits can maintain the integral morphology after being soaked in the electrolyte, proving its efficiency as an anti-corrosive layer (Supplementary Fig. 27). In addition, negative electrodes with MgF₂@PVDF, AlF₃@PVDF, and ZnF₂@PVDF layers obviously delivered lower nucleation overpotential and plating/stripping overpotential than other electrodes (Fig. 4b and Supplementary Fig. 28), benefiting from the superior lithiophilicity[50]. To further validate the passivation of the interface, the Li symmetric tests were conducted (Supplementary Fig. 29). Bare Cu electrode delivered limited stability (190 h), while

MFₓ@PVDF-Cu expectedly showed extended lifespans (Fig. 4e, f). Additionally, the voltage hysteresis of Li plating/stripping was largely reduced after the introduction of artificial passivation layers (Supplementary Fig. 30). Of note, the electrode with MgF₂@PVDF had the longest operation time of 1700 h and delivered low voltage hysteresis of 34.6, 44.2, 47.9, 62.8, 41.0, 31.9, 22.9, and 30.9 mV at 1.0, 2.0, 3.0, 5.0, 3.0, 2.0, 1.0, and 2.0 mA cm⁻², respectively. These artificial passivation layers are validated to regulate the nucleation and the subsequent Li plating/stripping processes effectively. The Li‖Cu and Li‖Li cells were enabled to deliver stable cyclability, large CE, and low voltage hysteresis, indicating a stable Li metal–liquid electrolyte interface, suppressed Li corrosion, and fast ion transport. Moreover, among artificial passivation layers with different fluorides, MgF₂@PVDF was validated as a representative and optimized artificial passivation layer for protecting Li metal negative electrode.

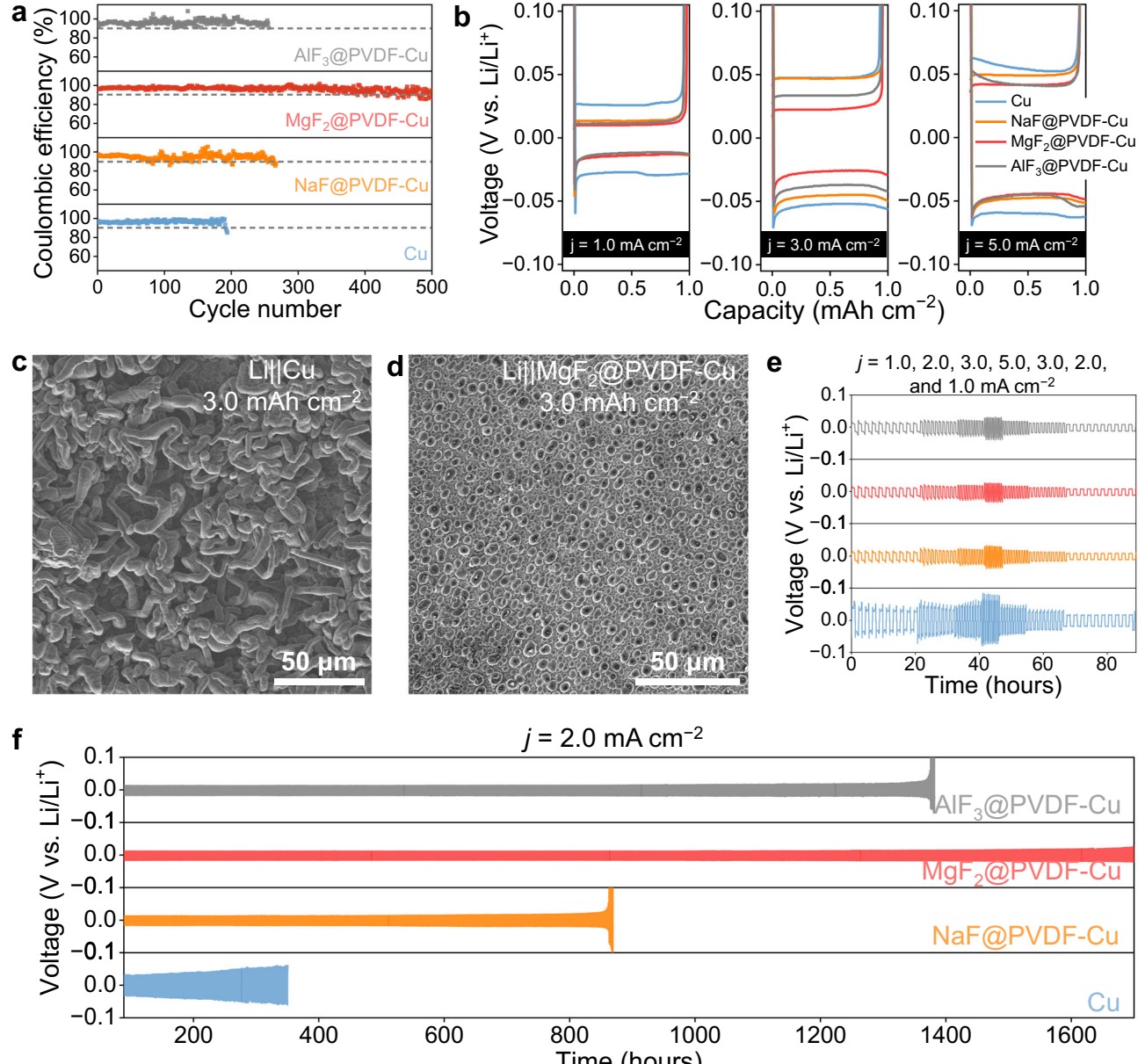

**Fig. 4 | Electrochemical evaluation of fluorides@PVDF artificial passivation layer in Li||Cu and Li||Li cell. a** CE results of Li||Cu cells with bare Cu, NaF@PVDF-Cu, MgF$_2$@PVDF-Cu, and AlF$_3$@PVDF-Cu electrodes at 1.0 mA cm$^{-2}$. **b** Voltage profiles of Li||Cu cells with bare Cu, NaF@PVDF-Cu, MgF$_2$@PVDF-Cu, and AlF$_3$@PVDF-Cu electrodes at 1.0, 3.0, and 5.0 mA cm$^{-2}$. **c, d** SEM images of Li deposits on bare Cu and MgF$_2$@PVDF-Cu, respectively. **e** Voltage profiles of Li||Li symmetric cells with bare Cu, NaF@PVDF-Cu, MgF$_2$@PVDF-Cu, and AlF$_3$@PVDF-Cu electrodes at 1.0, 2.0, 3.0, 5.0, 3.0, 2.0, and 1.0 mA cm$^{-2}$ in the 0–89 h. **f** Voltage profiles of Li||Li symmetric cells with bare Cu, NaF@PVDF-Cu, MgF$_2$@PVDF-Cu, and AlF$_3$@PVDF-Cu electrodes at 2.0 mA cm$^{-2}$ in the following 89–1700 h. The electrolyte was 1.0 M LiTFSI in DOL/DME with 1.0 wt% LiNO$_3$.

## Electrochemical evaluations of cells comprising Li metal negative electrodes and commercial positive electrodes

To verify the validity of such artificial passivation layers in real cells, the cell configuration paired with commercial LiFePO$_4$ positive electrodes was assembled and evaluated. Fig. 5a shows the rate capability of cells with and without the MgF$_2$@PVDF layer. The cell with MgF$_2$@PVDF delivered superior rate performance with capacities of 166, 164, 156, 142, 127, and 99 mAh g$^{-1}$ at 0.13, 0.26, 0.65, 1.3, 2.6, and 6.5 mA cm$^{-2}$, respectively. Notably, the MgF$_2$@PVDF layer enabled the cell to be stably operated even at an extremely fast charging/discharging rate of 13 mA cm$^{-2}$. In great contrast, the cell without the protection of an artificial passivation layer exhibited deteriorated rate performances, which can hardly work at large rates like 6.5 and 13 mA cm$^{-2}$. As seen in the voltage profiles at different current densities, the cell with

MgF$_2$@PVDF-Li obviously delivered lower overpotential and higher capacity than that of the cell with bare Li foil (Fig. 5b, c). The distinguishment of rate performances highly relates to the features of the interface. As protected by the MgF$_2$@PVDF layer, the continuous Li corrosion and growth of native SEI have been largely restrained. The stabilized interface promises fast interfacial kinetics, delivering satisfactory rate performances. However, the bare Li suffers a rapidly thickening SEI, making the diffusion and transport of ions become more and more sluggish. The prolonged cycling test also indicated the positive role of the MgF$_2$@PVDF layer in inhibiting Li corrosion and stabilizing the Li metal–liquid electrolyte interface (Fig. 5d, e). The Li|| LiFePO$_4$ cell with MgF$_2$@PVDF layer maintained ~80% of the initial capacity and CE over 96% after ~1500 cycles at 1.3 mA cm$^{-2}$. The feasibility of the MgF$_2$@PVDF layer was preliminarily validated in real

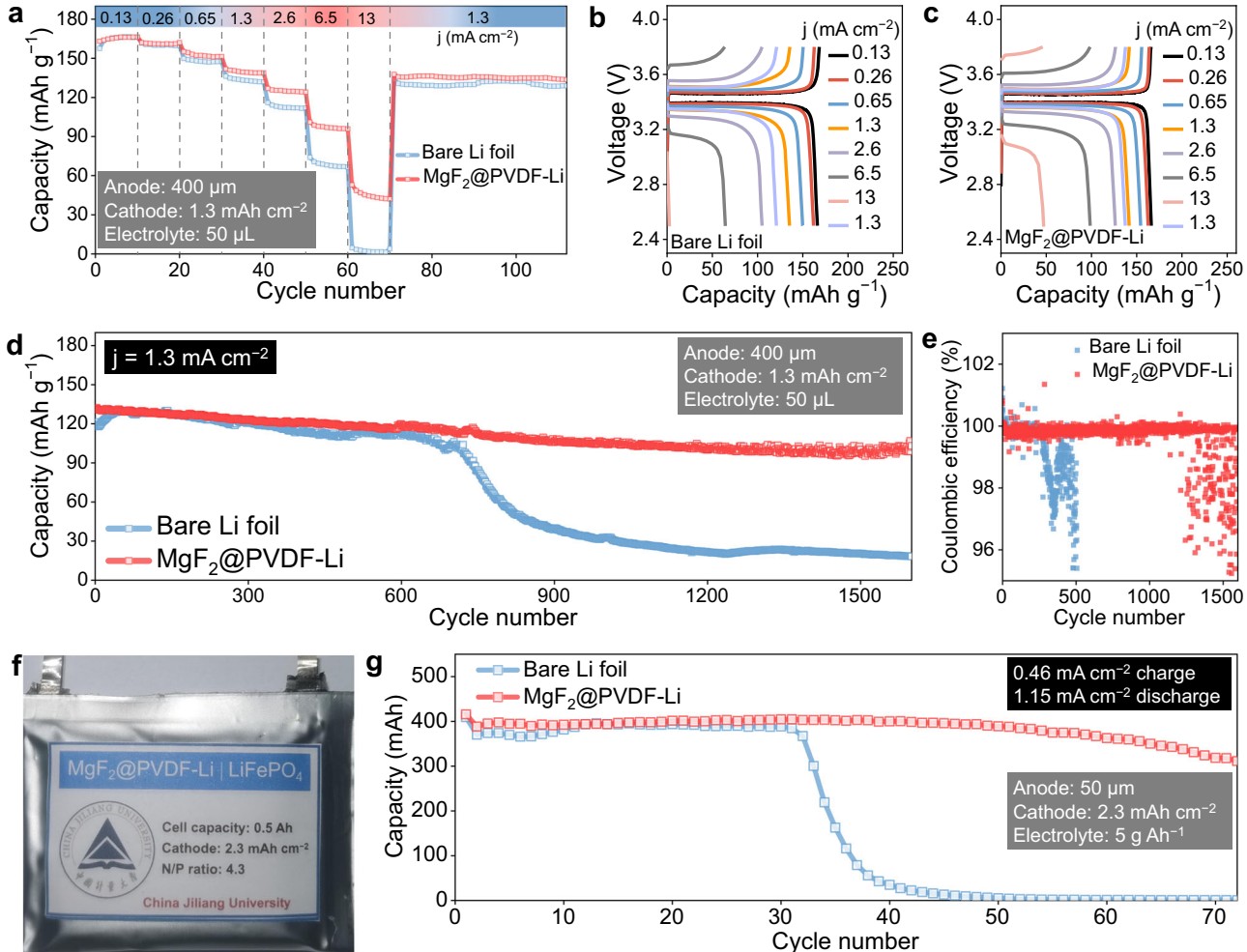

**Fig. 5 | Cell evaluations in pair with LiFePO₄ positive electrode. a** Rate capability of Li∥LiFePO₄ cells with and without a MgF₂@PVDF layer. **b**, **c** Voltage profiles of Li∥LiFePO₄ cells without and with a MgF₂@PVDF layer, respectively. **d**, **e** Prolonged cycling performance of Li∥LiFePO₄ cells with and without a MgF₂@PVDF layer and the corresponding CE. **f** Digital photo of a MgF₂@PVDF-Li∥LiFePO₄ pouch cell. **g** Cycling performances of Li∥LiFePO₄ pouch cells with and without a MgF₂@PVDF layer.

ampere-hour-scale pouch cells with high-loading LiFePO₄ (2.3 mAh cm⁻²) and NCM523 (3.2 mAh cm⁻²). Expectedly, the Li∥LiFePO₄ pouch cell with MgF₂@PVDF layer delivered twice better cycling stability than the control group, exhibiting a capacity retention of 77% after 70 cycles (Fig. 5f, g). The success can further be translated into NCM523-based cells (Supplementary Fig. 31). The evaluations of Li∥LiFePO₄ and Li∥NCM523 cells again validated the functions of such artificial passivation layers in suppressing Li corrosion and maintaining the stability of the interface. However, the practical application of such a strategy in real cells needs to be further optimized and developed.

In conclusion, Li corrosion behaviors associated with SEI dissolution have been quantitatively uncovered. Such dynamic evolutions at the Li metal−liquid electrolyte interface largely decide the stability and performances of Li metal batteries. A prototypical strategy is to introduce a protective layer to inhibit the SEI dissolution and Li corrosion by electrolyte. The matrix of such an artificial passivation layer should be stable against electrolyte, Li metal, and interfacial stress (e.g., low solubility, reductive stability, flexibility, and mechanical stability), while the filler inside the layer can promise uniform ion transport and Li deposition (e.g., ionic conductivity and lithiophilicity). Notably, the composite comprised of polymer and fluorides (e.g., MgF₂@PVDF) can serve as a typically ideal artificial passivation layer to reduce the corrosion of reactive negative electrodes like Li metal by 74%. The PVDF matrix enables the isolation of Li metal and native SEI

from electrolytes, while the fluorides react with Li to form corrosion-resistant Li alloy and LiF to suppress Li corrosion further. Additionally, the artificial passivation layer successfully facilitates ion transport and deposition, preventing dendrite growth and dead Li accumulation. Consequently, both coin-type and pouch cells exhibited superior cycling stability and remarkably extended lifespans. This work supplements the lack of fundamental recognition of Li corrosion and its correlation with SEI progression. The findings and strategy can further be extended to other battery systems using reactive metallic negative electrodes (e.g., Na, K, and Zn) and promote the development of corrosion science in battery research.

## Methods

### Fabrication of artificial passivation layers on Cu and Li foil

PVDF (battery grade, DoDoChem) and micron-sized metal fluorides (LiF, NaF, MgF₂, AlF₃, ZnF₂, LaF₃, and CaF₂, ~99.9%, ~100 μm, aladdin) were purchased and used after grinding. A slurry comprising fluorides, NMP (battery grade, DoDoChem), and PVDF (1:2:40 by weight) was prepared with a mixer. The slurry of artificial passivation layer was brushed onto the Cu or Li foil (~14 μL cm⁻²) and dried at 50 °C in a glove box or dry room. Notably, the mass loading of an artificial passivation layer on each metal foil was calculated to be ~1.1 mg cm⁻², and the content of fluorides in each layer was set to be ~30 wt% in this work. Li foil (a thickness of 0.4 mm and a diameter of 12 mm) and Li belt (a

thickness of 50 μm) with a high purity of 99.95% were purchased from China Energy Lithium Co., Ltd. Two kinds of battery-grade Cu were used, including a Cu foam (a thickness of 1.6 mm, purity of 99.8%) and a Cu foil (a thickness of 0.1 mm, purity of 99.9%), which were obtained from Jinghong New Energy Co., Ltd.

## Characterization

XRD patterns of samples in this work were conducted on an X'Pert Pro diffractometer deploying Cu $K_\alpha$ radiation ($\lambda = 0.15418$ nm). The morphology and microstructure of samples in this work were observed by SEM (FEI, Nova NanoSEM 450) and TEM (FEI, Talos-S). Elemental analysis was performed on an EDX spectrometer attached to TEM. Cryo-TEM was characterized by using a cryo-transfer holder of Gatan 698. Notably, a dilute solution of PVDF and fluoride was applied to reduce the thickness of the artificial passivation layer with the purpose of better visualizing the interface during TEM characterization. EELS was conducted by a FEI Titan G2 transmission electron microscope with an aberration corrector for a condenser lens operated at an accelerating voltage of 300 kV. XPS measurements were performed using an Al Kα monochromatic X-ray source (1486.6 eV, Axis Ultra DLD, Kratos). Infrared absorption spectra were measured at room temperature on an FTIR spectrometer (FT-IR, V80, Bruker Corporation).

## Battery assembling and evaluation

Typical 2032 coin-type cells were assembled to evaluate the effects of various $MF_x$@PVDF layers with an electrolyte amount of 50 μL. $MF_x$@PVDF-Cu electrodes were paired with Li foil to prepare Li||Cu cells. The symmetric cells were obtained by pre-depositing Li onto Cu in Li||Cu cells with an area capacity of 10.0 mAh cm$^{-2}$. The coin cells were tested in galvanostatic mode with Neware battery testing equipment (BTS-5V5/10/20 mA, Neware Technology Limited). For CE of Li||Cu cells, 1.0 mAh cm$^{-2}$ Li was plated into the $MF_x$@PVDF-Cu electrodes at a specific current density. Then the cell was charged to 1.5 V at the same current to strip out the plated Li. The efficiency was decided by the ratio between the amount of deposited and stripped Li. The symmetric cells were charged and discharged at certain current densities with a Li cycling capacity of 1.0 mAh cm$^{-2}$. The Li||LiFePO$_4$ cells were composed of LiFePO$_4$ positive electrodes and MgF$_2$@PVDF-Li or bare Li foil. LiFePO$_4$ positive electrode (7.6 mg cm$^{-2}$, ~1.3 mAh cm$^{-2}$) was prepared by coating the slurry of 80% LiFePO$_4$, 10% PVDF, and 10% Super P conducting carbon onto Al foil. The liquid electrolyte consisted of 1.0 M LiTFSI (99.95% trace metals basis, DoDochem) and 1.0 wt% LiNO$_3$ (99%, Alfa Aesar) dissolved in a mixture of DOL and DME (v/v = 1:1) was selected as the electrolyte and Celgard 2400 as the separator. EIS was measured with a CHI660E electrochemical workstation with a frequency ranging from 100 kHz to 0.1 Hz. Tafel plots were obtained from a cyclic voltammetry test in Li||Li cells with a scanning rate of 1.0 mV s$^{-1}$. The Tafel plot of the log current versus cell polarization was linearly fit over the voltage range from −25.0 to −50.0 mV. The value of exchange current density can be obtained from the intersection of the extrapolated linear part of the lg $j$ with the equilibrium potential ($\eta = 0$) line. The pouch cells were assembled by pairing 4.0 cm × 5.0 cm double-sided LiFePO$_4$ (content of positive electrode material: 80.0 wt%, mass loading: 16.9 mg cm$^{-2}$, density: 1.7 g cm$^{-3}$, Gushen) or NCM523 positive electrodes (content of positive electrode material: 94.0 wt%, mass loading: 19.7 mg cm$^{-2}$, density: 2.9 g cm$^{-3}$, Canard) with 4.0 cm × 5.0 cm × 50.0 μm double-sided Li foil copper inlay, employing the 5.0 cm × 6.0 cm separators. 1.0 M LiTFSI in DOL/DME (v/v = 1:1) with 1.0 wt% LiNO$_3$ and 1.0 M LiPF$_6$ in FEC/dimethyl carbonate (DMC) (v/v = 1:4) electrolytes were used in Li||LiFePO$_4$ and Li||NCM523 pouch cells, respectively. All the cells were tested in a battery testing chamber (BLC-300, Shanghai Biolab Equipment Co., Ltd.) with a setting temperature of 25 °C.

## Li corrosion and SEI dissolution tests

Li||Cu in typical 2032 coin-type cells were assembled to quantify the Li corrosion. A certain amount of Li (1.0 mAh cm$^{-2}$) was first plated onto the Cu foil. Next, the cell was stopped for a certain time (0, 5, 15, 50, 300, and 500 h), during which Li was corroded by the electrolyte. After that, Li was totally stripping. The amount of Li for SEI formation can be obtained from the cell without the rest treatment. Consequently, the capacity loss between Li plating and stripping can indicate the degree of Li corrosion. Li||Cu cell was also used to determine the SEI dissolution rates in different electrolytes. The cells were cycled from 0.05 to 2.0 V vs. Li/Li$^+$ for 5 cycles to produce SEI on the electrode before pausing for 50 h. The same procedure was repeated with a pause time of 30, 15, and 5 h. The extra reductive capacity after the pauses indicates the degree of SEI dissolution. The electrolytes include 1.0 M LiTFSI in DOL/DME (v/v = 1:1), LiTFSI in DOL/DME with 1.0 wt% LiNO$_3$ (v/v = 1:1), LiPF$_6$ in EC/DMC/DEC (v/v/v = 1:1:1), LiPF$_6$ in EC/EMC/DEC with 1.0 wt% FEC (v/v/v = 1:1:1), and LiPF$_6$ in FEC/DMC (v/v = 1:4). The electrolyte amount used in cells for Li corrosion and SEI dissolution tests was 50 μL.

## Data availability

All data are available in the main text or the Supplementary Information, which can also be available from the corresponding authors upon request. Source data are provided in this paper.

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

## Acknowledgements

The authors acknowledge the funding support of the National Natural Science Foundation of China (grant No. 52103342, C.B.J.; grant No. 22209032, O.W.S.; grant Nos. 52225208, U21A20174, and 51972285, X.Y.T.), and China Jiliang University Research Fund Program for Young Scholars (grant No. 01101-221040, C.B.J.). We also acknowledge Dr. Zhenhua Zhang of Hangzhou Dianzi University for helping us conduct the EELS test and analysis.

## Author contributions

C.B.J., O.W.S., and X.Y.T. conceived and designed the experiments. C.B.J., O.W.S., J.L.Z., Z.J.J., Q.Y.S., L.H.L., Y.Y.H., and G.X.L. directed the project and carried out the experiments. C.B.J., O.W.S., G.Y.W., H.Y.L., and X.Y.T. cowrote the paper. All authors discussed the results and commented on the paper.

## Competing interests
The authors declare no competing interests.
