## [Peer Review File · Nature Communications]

Reviewers' comments:

Reviewer #1 (Remarks to the Author):

The received manuscript is dedicated to interesting and well-timed topic of Li metal protection related to the emerging Li metal battery field. Quality of manuscript is quite good. Results are interesting for the LMB field.

However, Li protection (including by layers based on "metal fluorides-PVdF" system) is well known approach 10.34133/energymatadv.0010, 10.1002/inf2.12189, 10.1002/batt.20200225).

Therefore, since the manuscript does not bring disruptively new knowledge to the battery community it should be rejected from publishing in Nature Communications. Hope, some of my comments below will help to authors to improve the manuscript quality:

All text: Authors several times refer to half and full coin cell type. This terminology is rather related to LIB topics when all Li-metal cells are considered as half-cells. Therefore, authors should revise the cell terminology in whole manuscript.

Figure 1e: I have my doubts that galvanic (electrochemical) corrosion is possible when corrosion spot of Li metal does not have direct electrical contact with inert current collector – typical situation for Li metal foil anodes with thickness more than few microns or current-collector less Li metal anodes.

Line 98. Decreasing of coulombic efficiency can be attributed to many factors starting from increase of impedance. Please discuss it.

Line 85-116: In principle, it is well-known Li metal tends to passivation and formation of robust and stable for years SEI layers. Therefore, authors should demonstrate some SEM images of the Li deposits to prove the continuous dissolution of SEI. Also, I suggest adding of EIS spectra to analyse the SEI evolution.

Line 138: Please explain why MgF₂ filler was selected and how was optimized its content in protection layer.

Figure 3a: Please report data for PVdF-coated Li metal for comparison purpose.

Lines 183-215: Authors should discuss why MgF₂ coating works better. Perhaps, due to formation of Li-Mg alloy?

Lines 257-263. Please specify grade, source, thickness, purity etc. of used Li and Co foils. Please specify particle size of all used inorganic fillers.

Line 275-300. Please specify more details about CAMs, loading (mg/cm²) and density (g/cc) of LFP and NMC based cathodes.

Figure 5f. Please add a digital photo of the pouch cell.

Line 243-254: Conclusions are not fully supported by presented experimental data. For example, scenarios of Li dendrites suppression, SEI composition/thickness modification and formation of Li-Mg alloy when used "MgF₂-PVdF" protection film are not discussed. Moreover, authors did not prove that Li metal is deposited under protection layer (Figure. S15).

SI, Figure S1. Please specify all experimental conditions (electrochemical system, current density, temperature etc.).

SI, Line 20: "Cell potential" is not correct term. Strictly speaking, cell voltage (correct term) is a difference of electrode potentials.

SI, Line 20-26: In my opinion, this statement is unsupported claim until authors will demonstrate that SEI layer was really dissolved, and Li metal remained intact.

Reviewer #2 (Remarks to the Author):

The authors devise a M(x)F(y)@PVDF coating to reduce "corrosion" on lithium metal anodes and find improved cycling capability. Cryo-TEM and other techniques are used to characterize the SEI layers.

I have major reservations about this paper and recommend a very major revision before it is reconsidered for publication.

(1) The word "corrosion" is used numerous times, yet it is unclear what precisely that means. Corrosion is a very well studied subject, and numerous categories of corrosion have been determined.

See, e.g., <https://waverleybrownall.co.uk/blog/5-common-types-corrosion/>

So when the authors say "corrosion," which category is it? What is the overpotential involved? Corrosion usually occurs at an overpotential.

If the authors only mean "lithium has reacted/degraded" without focusing on corrosion mechanisms, they should consider just using "aging" or "self-discharge."

(2) Which electrolyte is used in Fig. 1a-d? Several papers, e.g., *Adv. Energy Mater.* 10, 2000017, (2020), and *ACS Appl. Energy Mater.* 4, 7589 (2021), have shown that the amount of "corrosion" or self-discharge depends on amount of Li deposited. How do those papers compare with the authors' results (Fig. 1f, Fig. 3a)?

(3) The abstract and conclusion stress that dissolution of the SEI is elucidated as a main cause of "corrosion." The Introduction even claims that the manuscript has proven this point. But there is absolutely no characterization showing that the SEI has dissolved. For example, detection of SEI fragments in the electrolyte would constitute "dissolution." Voltage-time curves (Fig. S1) do not. It is indisputable that in this aspect, the data does not support the authors' conclusion.

Does Mg(2+) dissolve into the electrolyte as well?

(4) Without having "proven" their key hypothesis, it is unclear what the main point of the paper is, apart from the fact that the authors have a new coating that improves cycling (Fig. 5d, Fig. 5g). They

cite 96.2% capacity after 500 cycles. What is the coulomb efficiency (CE) ? How does it compare with the numerous papers in the literature which use coatings to improve the CE? I guess the authors need to show the novelty in this paper and tell the reader why it is distinct from so many related papers in the literature.

Reviewer #3 (Remarks to the Author):

The authors report how a fluoride/PVDF coating can help suppress corrosion of Li. The report that coatings can help prevent corrosion of Li is notable and remains an area to explore further. Especially as a method to augment electrolyte engineering. This alone is an interesting topic, but several points are necessary to address below to make this paper of suitable quality in this reviewer's opinion.

Certain portions of the writing are over exaggerated:

"This work unveils the origin of Li corrosion" and "the direct correlation between dynamic Li corrosion behaviors and SEI progression presently remains unknown" - the connection of SEI progression with corrosion was investigated closely in reference 30, as well as others - doi: 10.1021/jacs.0c10258, 10.1002/aenm.202202012,

Certain areas need more support or explanation:

"Notably, the SEI dissolution can be detected in various ether and ester electrolytes" - is there a reference or measurement to support this claim? dissolution of the SEI is proposed to be a major cause of corrosion and a key finding in the abstract (mentioned twice), but it isn't tracked at all in the manuscript.

Fluorides have low ionic conductivity. Why are the authors referring to a fluoride based SEI as having high conductivity? Furthermore, although the literature commonly claims high ionic conductivity of the SEI is needed for better cyclability, recent authoritative work that drew correlations across many electrolytes have suggested this is not a leading predictor of high CE: doi: 10.1038/s41560-021-

00910-w (Nature Energy), 10.1021/jacs.2c08182 (JACS), 10.1038/s41560-022-01144-0 (Nature Energy)

The SEM and TEM images of Li and the polymer coating are not aligned. Why in the SEM images is the polymer thickness 25 microns thick, but in TEM the polymer thickness is believed to be only 10 nm thick. EELS is the better method to assign chemistry especially when many materials appear amorphous or are not aligned clearly to lattice fringes.

For the Tafel plot analysis. The low over potential slope, rather than linear extrapolation from high over potential regions, should be used to calculate the exchange current density. Many reports in the literature also make this mistake. This may fix the discrepancy between the EIS and Tafel data, though. Right now Fig. 3b,c are not in agreement as would be expected for such an analysis. Furthermore, why is the exchange current density increasing with time more for MgF/PVDF coating? Usually the resistance of the interface increases with rest time because the SEI grows.

For the battery data. I don't believe the loading is reported for the coin cell data in Fig. 5 a-c, making the C-rates and capacity retention in the coin cells meaningless at this stage. It's notable that while 10C and 1500 cycles are advertised here, 0.2 C is used for the pouch cell that only reached 70 cyc with a high N/P of 4.3. As mentioned in the beginning of this review, the clear demonstration that such coatings can prevent corrosion is itself valuable, but exaggeration of battery cyclability isn't helpful.

Point-by-point response to the referee questions

Reviewer #1 (Remarks to the Author):

The received manuscript is dedicated to interesting and well-timed topic of Li metal protection related to the emerging Li metal battery field. Quality of manuscript is quite good. Results are interesting for the LMB field. However, Li protection (including by layers based on “metal fluorides-PVdF” system) is well known approach 10.34133/energymatadv.0010, 10.1002/inf2.12189, 10.1002/batt.202000225). Therefore, since the manuscript does not bring disruptively new knowledge to the battery community it should be rejected from publishing in Nature Communications. Hope, some my comments below will help to authors to improve the manuscript quality:

Response:

We are grateful to the reviewer for nice reviews of our paper. According to the reviewer’s suggestions, we addressed all the detailed comments below and revised our manuscript, especially for the supplement of the novelty and advances of our work compared to the previous reports. Li protection is of great significance for the durable operation of Li metal batteries, which have been reported in many works. However, **the main focus of most publications on Li protection is to use artificial SEI to enhance the interfacial stability and suppress the dendrite growth**, while the direct recognition on dynamic Li corrosion during battery operation and storage still needs to be enriched.

Very recently, some researches also show the awareness of the SEI dissolution issue^{1,2}, while its quantitative correlation with the Li corrosion and performance deterioration of batteries remains to be further explored. **The key point of our work is trying to track the SEI dissolution behaviors, and uncover the correlation between SEI dissolution and Li corrosion by electrolyte, being an emerging and practical topic in the community of Li metal battery.** In the revised manuscript, we have directly validated the SEI dissolution and Li corrosion by visualized SEM and cryo-TEM images, including the morphologic, structural and chemical changes of SEI and Li before and after soaking in the electrolyte. Besides, the shielding layer comprising of PVDF and fluorides serves as a prototypical solution to the SEI dissolution triggered Li corrosion, which can be further extended to a large family on basis of the findings in our work. The distinguishments between our work and several typical reports on

artificial SEI and other Li protection strategies has been summarized. And the mentioned papers have been included and cited. All the changes in the revised manuscript were highlighted with yellow, which can also be found below.

“*Very recently, Cui and Zhang et al. have validated the SEI dissolution in alkali metal battery systems, which has a significant effect on the cyclability of batteries^{31,32}. However, the direct correlation between dynamic Li corrosion behaviors and SEI progression (e.g., SEI dissolution and reformation) still needs more expeditions (Supplementary Table S1)*”. (Page 3, Line 3–7)

“6 Li, J. et al. Strategies to anode protection in lithium metal battery: A review. *InfoMat* 3, 1333-1363, doi:10.1002/inf2.12189 (2021).

7 Kang, D., Xiao, M. & Lemmon, J. P. Artificial solid-electrolyte interphase for lithium metal batteries. *Batteries & Supercaps* 4, 445-455, doi:10.1002/batt.202000225 (2020).

9 Bi, C.-X. et al. Protecting lithium metal anodes in lithium–sulfur batteries: A review. *Energy Mater. Adv.* 4, 0010, doi:10.34133/energymatadv.0010 (2023)”. (Supplementary information, Page S37, Line 13–16, and Line 19, 20)

Table R1. The advances in researches on Li corrosion and protection.

Main content	Li protection strategy	Ref
The fast galvanic Li corrosion involving a Kirkendall-type mechanism	Passivation with a good electrical insulation and low electrolyte permeability	3
The galvanic corrosion of lithium-powder-based electrodes	Isolating copper from the electrolyte by additives, interfacial engineering, and physical vapor deposition techniques	4
The chemical corrosion of Li and continuous growth of the SEI during calendar ageing	Minimizing the surface area of Li and the rate of SEI growth, and reusing existing SEIs from previous cycles	5
The relationship between aging and cycling on self-discharge of Li metal batteries	Increasing the capacity of plated lithium, tuning electrolyte chemistry, designing rest protocols, and using lithiophilic coatings	6
A review on strategies to Li	Suppressing the dendrite growth and	7

protection	stabilizing interphase	
A review on the artificial SEI for protecting Li	Reviewing the artificial SEI for protecting Li from corrosion by air and water	8
Chemically induced activity recovery of isolated lithium	A block copolymer coating to induce the reconnection of isolated Li	9
A review on reducing the Li corrosion by polysulfide	Guiding uniform Li plating/stripping, and reducing the concentration and reaction activity of polysulfide	10
The relation between chemical corrosion and Li depositing morphology	Controlling the stacking pressure during Li plating to reduce the porosity of Li deposits	11
The dynamic galvanic corrosion mechanism of Li metal during Li stripping process	A high-rate Li stripping protocol	12
The SEI dissolution during rest at open circuit has been quantified	Tuning the SEI composition and the physical and chemical properties of the electrolyte	2
The correlation between Li corrosion and dynamic SEI dissolution	A composite shield to inhibit the dissolution of native SEI and construct a superiorly stable interface	This work

Obviously, **our work provides in-depth understanding on the correlation between Li corrosion and SEI dissolution**, which sheds certain new light on stabilizing the Li metal anode for prolonged and stable cycling. We believe that our manuscript after addressing all the valuable comments and suggestions raised by the reviewers could be considered for publication in Nature Communications.

Comment 1: All text: Authors several times refer to half and full coin cell type. This terminology is rather related to LIB topics when all Li-metal cells are considered as half-cells. Therefore, authors should revise the cell terminology in whole manuscript.

Response:

Thank you for the detailed suggestion. The half-cell used in the manuscript supposedly refers to Li | Cu and Li | Li cell. We've corrected such terminologies of half and full cell by clearly describing the cell configurations of Li | Cu, Li | Li, Li | LiFePO₄, and Li | NCM523 cell. The corresponding corrections were highlighted with yellow in the revised manuscript.

Comment 2: Figure 1e: I have my doubts that galvanic (electrochemical) corrosion is possible when corrosion spot of Li metal does not have direct electrical contact with inert current collector – typical situation for Li metal foil anodes with thickness more than few microns or current-collector less Li metal anodes.

Response:

Thank the reviewer for raising such a professional question. We agree that due to the high-capacity Li deposition with a dense morphology, the Cu surface is rarely exposed to electrolyte, suppressing the galvanic corrosion. However, during the subsequent Li stripping, fresh Cu surface will be progressively exposed, again initiating the galvanic corrosion¹². Besides, Li metal anode will become loose and porous after repeated cycling, where the dynamic galvanic corrosion potentially occurs with the penetration of electrolyte. Moreover, in addition to the galvanic corrosion, chemical corrosion (*i.e.*, pitting corrosion) still undergoes during battery storage and cycling at both low and high Li deposition capacity (*e.g.*, 0.5 and 3.0 mAh cm⁻², **Fig. R1**). The figure and relevant discussion have been added in the revised manuscript and supplementary information, which can also be seen below.

*“Such Li deposits suffering chemical and/or electrochemical corrosion (*i.e.*, pitting corrosion and galvanic corrosion) can be found in typical ether and ester electrolytes (Fig. 1e, and Supplementary Figs. S1 and S2)”. (Page 3, Line 33, and Page 4, Line 1–3)*

“Even at a large Li deposition capacity where Cu was totally covered by Li, Li corrosion still can be observed in the Li deposits, which can be ascribed to the chemical corrosion by the electrolyte. Additionally, during the subsequent Li stripping, fresh Cu surface covered with Li deposits will be progressively exposed, again initiating the galvanic corrosion¹. Besides, Li metal anode will become loose and porous after repeated cycling, where the dynamic galvanic corrosion potentially occurs with the penetration

of electrolyte". (Supplementary information, Page S3, Line 4–10)

Fig. R1 | Morphologies of Li deposits after soaking in 1.0 M LiTFSI-DOL/DME with 1.0 wt% LiNO₃. a–d SEM images of Li deposits formed at 1.0 mA cm⁻² with a capacity of 3.0 mAh cm⁻².

Comment 3: Line 98. Decreasing of coulombic efficiency can be attributed to many factors starting from increase of impedance. Please discuss it.

Response:

We agree with the reviewer that many factors may account for the deterioration of CE. In Fig. 1f, Li | Cu cells were assembled and tested under almost the same conditions, except the only difference in the resting time after each Li plating (0, 5, 15, and 50 hours, **Fig. R2a**). During the duration of rest, SEI growth and Li corrosion continue to contribute to the accumulation of dead Li and the increase of cell impedance (**Fig. R2b**), which are responsible for the decrease of CE. We have added relevant discussion in the revised manuscript, which can also be found below.

“The continuous chemical and electrochemical Li corrosion proceeds originating from the inferiority and progression of initially formed SEI, which may contribute to the accumulation of dead Li and increase of cell impedance. Consequently, cells after durations of rest delivered enlarged capacity loss of Li deposits ($> 240 \mu\text{Ah}$ for 50 hours) and deteriorated CE ($< 80\%$)”. (Page 4, Line 9–13)

Fig. R2 | Voltage profiles and EIS curves of Li | Cu cells with different resting treatments. **a** Voltage profiles of Li | Cu with different resting time (1.0 M LiTFSI-DOL/DME with 1.0 wt% LiNO₃, 1.0 mA cm⁻², 1.0 mAh cm⁻²). **b** EIS curves of Li | Li@Cu after different resting time in 1.0 M LiTFSI-DOL/DME with 1.0 wt% LiNO₃. Li of 0.5 mAh cm⁻² was plated onto the Cu in Li | Cu cell at 1.0 mA cm⁻².

As shown in Fig. R2b, the impedance of Li | Li@Cu cell enlarged with the increase of resting time due to the deterioration of interface and accumulation of dead Li, which may cause the incomplete stripping of Li, and consequently the decrease of CE.

Comment 4: Line 85-116: In principle, it is well-known Li metal tends to passivation and formation of robust and stable for years SEI layers. Therefore, authors should demonstrate some SEM images of the Li deposits to prove the continuous dissolution of SEI. Also, I suggest adding of EIS spectra to analyse the SEI evolution.

Response:

Thanks for the professional and valuable suggestion. SEI experiences complex dynamic evolutions during battery operation, due to the effect from electrolyte solvation, volumetric changes, etc. SEI reaches a dynamically stable state after balancing the SEI

dissolution/breakage and reformation/repair processes. Referring to the reviewer's comment, we have supplemented SEM, cryo-TEM images and EIS to prove the SEI dissolution and evolution. Relevant discussion and figures have been added in the revised manuscript, which can also be found below.

As shown in Fig. R3, Li deposits in typical ether and ester electrolytes exhibit obvious corrosion after soaking in the electrolyte, which is related to the SEI dissolution. In the ether electrolyte (1.0 M LiTFSI-DOL/DME with 1.0 wt% LiNO₃), spherical Li has been corroded with obvious voids. In terms of ester electrolyte (1.0 M LiPF₆-EC/EMC/DEC with 1.0 wt% FEC), the surface of the dendritic Li becomes rough and porous after soaking.

Fig. R3 | Morphologies of Li deposits before and after soaking in the electrolytes. **a–d** SEM images of Li deposits in 1.0 M LiTFSI-DOL/DME with 1.0 wt% LiNO₃ before and after soaking. **e–h** SEM images of Li deposits in 1.0 M LiPF₆-EC/EMC/DEC with 1.0 wt% FEC before and after soaking. The Li was deposited at 1.0 mA cm⁻² with a depositing capacity of 0.5 mAh cm⁻².

To clearly validate the SEI dissolution behavior, SEM images of SEI on Cu before and after soaking were collected (Fig. R4a, b). The initial SEI residual on Cu was dark and dense. After soaking, the SEI residual becomes loose and bright, which indicates the SEI dissolution in the electrolyte. The similar finding has also been detected in the cryo-TEM images (Fig. R4c, d).

Fig. R4 | Morphologies of SEI before and after soaking in the electrolytes. **a, b** SEM images of SEI on the Cu foil formed in 1.0 M LiTFSI-DOL/DME with 1.0 wt% LiNO₃ before and after soaking. **c, d** Cryo-TEM images of SEI on the Cu foil formed in 1.0 M LiTFSI-DOL/DME with 1.0 wt% LiNO₃ before and after soaking. The SEI was produced after one plating/stripping process of a Li | Cu cell at 1.0 mA cm⁻² and 0.5 mAh cm⁻².

We have also supplemented the EIS spectra of Li | Li@Cu as a function of time in typical ether (1.0 M LiTFSI-DOL/DME with 1.0 wt% LiNO₃) and ester (1.0 M LiPF₆-EC/EMC/DEC with 1.0 wt% FEC) electrolytes. With the increase of resting time, the impedance of SEI gradually enlarges (Fig. R5a–c). Conclusively, during the cycling and storage of cells, SEI experiences dynamic evolutions, including repeated breakage/dissolution and repair/reformation, which result in the continuous loss of active Li source and the net growth of SEI (Fig. R5d).

Fig. R5 | EIS results of Li | Li@Cu and SEI evolution. **a** EIS curves of Li | Li@Cu after different resting time in 1.0 M LiTFSI-DOL/DME with 1.0 wt% LiNO₃. **b** EIS curves of Li | Li@Cu after different resting time in 1.0 M LiPF₆-EC/EMC/DEC with 1.0 wt% FEC. Li of 0.5 mAh cm⁻² was plated onto the Cu in Li | Cu cell at 1.0 mA cm⁻². **c** Impedance of SEI derived by fitting the EIS results. Inset is the equivalent circuit. **d** A schematic illustration of the SEI evolution, and the correlation with the net growth of SEI and continuous Li corrosion.

“Moreover, the SEI dissolution was directly validated by detecting the morphological, structural and chemical changes of SEI before and after soaking in the electrolyte. The SEI suffered obvious shrinkage after soaking in the electrolyte (Fig. 1g, h, and Supplementary Figs. S4–6). Additionally, the SEI after soaking contained abundant Li salts and limited polymeric composition, indicating the higher solubility of organic components (Fig. 1i and Supplementary Fig. S7). To confirm such hypothesis, the dissolved SEI compositions in the DOL/DME were collected and studied, which was found to be comprised of a large number of amorphous polymeric substances and a few crystalline Li salts (Fig. 1j)”. (Page 4, Line 22–31)

“Such progress further induces the net growth of flocculent SEI, and seriously deteriorates the lifespans of Li metal batteries (Supplementary Figs. S3d and S8)³²”.

(Page 4, Line 33, 34, and Page 5, Line 1)

“With the increase of resting time, the impedance of SEI gradually enlarges, indicating the dynamic evolutions of SEI. The repeated breakage/dissolution and repair/reformation of SEI will result in the continuous loss of active Li source and the net growth of SEI”. (Supplementary information, Page S9, Line 5–8)

Comment 5: Line 138: Please explain why MgF₂ filler was selected and how was optimized its content in protection layer.

Response:

Thank you for your valuable question. MgF₂ was selected due to its potential in regulating the uniform Li deposition and constructing a stable interface. In detail, Mg can sever as a superior lithiphilic seed (Fig. R6a), while F anions can contribute to a LiF-rich stable SEI. Besides, MgF₂ compared to other fluorides delivered superior performances during Li | Li and Li | Cu test (Figs. R7 and R8). The function of lithiophilic seeds (Mg, Al, Zn, La, Ca, *etc.*) and fluoridation strategy have been systematically studied and validated in our previous works¹³⁻¹⁷.

Although MgF₂ plays a significant role in stabilizing Li metal anode, the introduction of such electrochemically inert fillers will reduce the energy density of cells and increase the cell impedance. On basis of the previous researches, and a representative CE evaluation (Fig. R6b), the optimized content of metal in this work was decided to be ~30 wt% of the composite shielding layer, which delivered the largest average CE. We have supplemented discussion on such concerns in the revised manuscript, which can also be seen below.

“The embedded metal fluorides promise the formation of LiF and lithiophilic alloy (e.g., Li-Mg) at the interface, which promote the superior ion transportation and uniform Li deposition⁴²”. (Page 5, Line 12–14)

“MgF₂ as a typical low-solubility and lithiophilic fluoride was selected as a prototype during the investigation of the shield (Supplementary Fig. S13)⁴². Note that the optimized content of MgF₂ was decided to be 30 wt% (Supplementary Fig. S14)”. (Page 5, Line 17–20)

Fig. R6 | Voltage profiles and CE of Li | Cu cells using different shields. a Voltage profiles of Li | Cu cells with different shields. **b** CE results of Li | Cu cells with shields of different content of MgF₂. The cells were tested at 1.0 mA cm⁻² and 1.0 mAh cm⁻². The electrolyte used was 1.0 M LiTFSI-DOL/DME with 1.0 wt% LiNO₃.

Fig. R7 | CE results of various electrodes at 1.0 mA cm⁻². a Bare Cu. **b** PVDF-Cu. **c** LiF@PVDF-Cu. **d** NaF@PVDF-Cu. **e** MgF₂@PVDF-Cu. **f** AlF₃@PVDF-Cu. **g** ZnF₂@PVDF-Cu. **h** CaF₂@PVDF-Cu. **i** LaF₃@PVDF-Cu. The electrolyte used was 1.0 M LiTFSI-DOL/DME with 1.0 wt% LiNO₃.

Fig. R8 | Voltage profiles of symmetric cells at 1, 2, 3, 5, 3, 2, 1, and 2 mA cm⁻². a Bare Cu. **b** PVDF-Cu. **c** LiF@PVDF-Cu. **d** NaF@PVDF-Cu. **e** MgF₂@PVDF-Cu. **f** AlF₃@PVDF-Cu. **g** ZnF₂@PVDF-Cu. **h** CaF₂@PVDF-Cu. **i** LaF₃@PVDF-Cu. The electrolyte used was 1.0 M LiTFSI-DOL/DME with 1.0 wt% LiNO₃.

Comment 6: Figure 3a: Please report data for PVdF-coated Li metal for comparison purpose.

Response:

Thanks for reminding us to provide the data for the control group. We've supplemented the data and discussion for PVDF coated Li in the revised manuscript, which can also be seen below (Fig. R9).

“In addition, the passivation of bare PVDF shield is weaker than that of MgF₂@PVDF, delivering higher capacity loss and lower CE, which validates the synergistic effect of PVDF and MgF₂ in Li protection”. (Page 6, Line 12–14)

Fig. R9 | Corrosion related capacity loss of Li and the corresponding CE as a function of resting time with the adoption of MgF₂@PVDF or PVDF shield. The electrolyte used was 1.0 M LiTFSI-DOL/DME with 1.0 wt% LiNO₃.

Comment 7: Lines 183-215: Authors should discuss why MgF₂ coating works better. Perhaps, due to formation of Li-Mg alloy?

Response:

Thank you for your suggestion. The shield with MgF₂ works better, since Mg is superiorly lithiophilic material compared to other metals in fluorides (Fig. R6a). The formation of Li-Mg alloy is thought to induce the uniform deposition and stable interface^{18,19}, which can to some extent contribute to suppressing Li corrosion. We’ve added discussion on this question in the revised manuscript, which were also presented as follows.

“The embedded metal fluorides promise the formation of LiF and lithiophilic alloy (e.g., Li-Mg) at the interface, which promote the superior ion transportation and uniform Li deposition⁴¹”. (Page 5, Line 12–14)

“Such improvement of CE potentially relates to the uniform Li deposition and reduced corrosion guided by Li-Mg alloy and LiF at the interface⁴⁷⁻⁵⁰”. (Page 7, Line 22, 23)

⁴¹ Wang, S. H. et al. Tuning wettability of molten lithium via a chemical strategy for lithium metal anodes. *Nat. Commun.* 10, 4930 (2019).

⁴² Kim, M. H. et al. Design principles for fluorinated interphase evolution via conversion-type alloying processes for anticorrosive lithium metal anodes. *Nano Lett.*

23, 3582-3591 (2023)”. (Page 15, Line 8–12)

“47 Kong, L. L. et al. Lithium–magnesium alloy as a stable anode for lithium–sulfur battery. *Adv. Funct. Mater.* 29, 1808756 (2019)”. (Page 15, Line 24, 25)

Comment 8: Lines 257-263. Please specify grade, source, thickness, purity etc. of used Li and Co foils. Please specify particle size of all used inorganic fillers.

Response:

Thanks for commenting on the details of materials. We have added such information in the experimental section of the revised manuscript, which can also be found below.

“PVDF (battery grade, DoDoChem) and metal fluorides (LiF, NaF, MgF₂, AlF₃, ZnF₂, LaF₃, and CaF₂, ~99.9%, ~100 μm, aladdin) were purchased and used after grinding”.

(Page 10, Line 3–5)

“Li foil (a thickness of 0.4 mm and a diameter of 12 mm) and Li belt (a thickness of 50 μm) with a high purity of 99.95% were purchased from China Energy Lithium Co., Ltd. Two kinds of battery-grade Cu were used, including a Cu foam (a thickness of 1.6 mm, purity of 99.8%) and a Cu foil (a thickness of 0.1 mm, purity of 99.9%), which were obtained from Jinghong New Energy Co., Ltd”. (Page 10, Line 9–13)

Comment 9: Line 275-300. Please specify mode details about CAMs, loading (mg/cm²) and density (g/cc) of LFP and NMC based cathodes.

Response:

Thank you for your kind reminding. We have added the parameters of LFP and NCM in the experimental section of the revised manuscript, which can also be found below (Table R2).

“LiFePO₄ cathode (7.6 mg cm⁻², ~1.3 mAh cm⁻²) was prepared by coating the slurry of 80% LiFePO₄, 10% PVDF and 10% Super P conducting carbon onto Al foil”. (Page

11, Line 8–10)

“The pouch cells were assembled by pairing 4.0 cm×5.0 cm double-sided LiFePO₄ (content of cathode: 80.0 wt%, mass loading: 16.9 mg cm⁻², density: 1.7 g cm⁻³, Gushen)

or NCM523 cathodes (content of cathode: 94.0 wt%, mass loading: 19.7 mg cm⁻², density: 2.9 g cm⁻³, Canard) with 4.0 cm×5.0 cm×50.0 μm double-sided Li foil copper inlay”. (Page 11, Line 20–24)

Table R2. Parameters for cathodes used in the pouch cells.

Items	LFP	NMC
Areal capacity (mAh cm ⁻²)	2.3	3.1
Mass loading (mg cm ⁻²)	16.9	19.7
Content of active material (wt%)	80.0	94.0
Density (g cm ⁻³)	1.7	2.9

Comment 10: Figure 5f. Please add a digital photo of the pouch cell.

Response:

We’ve added a digital photo of the pouch cell in Fig. 5g, which can also be seen below (Fig. R10).

Fig. R10 | A digital photo of the MgF₂@PVDF-Li | LiFePO₄ pouch cell.

“*Inset is the digital photo of a pouch cell*”. (Page 21, Line 7, 8)

Comment 11: Line 243-254: Conclusions are not fully supported by presented experimental data. For example, scenarios of Li dendrites suppression, SEI composition/thickness modification and formation of Li-Mg alloy when used “MgF₂-

PVdF” protection film are not discussed. Moreover, authors did not prove that Li metal is deposited under protection layer (Figure. S15).

Response:

Thank the reviewer for such valuable suggestions. We have supplemented the necessary data to support the conclusions. As shown in Fig. R11, the uniform Li deposition behavior (0.5 mAh cm^{-2}) under the $\text{MgF}_2@\text{PVDF}$ shield has been validated, where a layer of spherical Li can be found. At a larger Li deposition capacity of 3.0 mAh cm^{-2} , dendrite-free Li deposition can still be maintained underneath the shielding layer, while the deposits on bare Cu foil were mossy Li dendrites (Fig. R12). Moreover, we have added comparison and discussion on SEI of Li deposits formed on bare Cu foil and underneath the $\text{MgF}_2@\text{PVDF}$ shield. With the adoption of a shield, a double layered SEI comprised of a polymer layer and an inorganic-rich layer was constructed, where Li-Mg alloys and various inorganic Li salts can be found (Fig. R13a–d). Moreover, electron energy loss spectroscopy (EELS) mapping also indicated the tight adhesion and uniform distribution of shield on the surface of Li deposit (Fig. R13e). In sharp contrast, the Li deposits on bare Cu grid after soaking showed obvious corroded morphology (Fig. R14a). Additionally, SEI showed a large thickness and contained randomly distributed crystals of Li_2O (Fig. R14b). The supplemented contents in the revised manuscript were also presented below.

“In contrast, the SEI of Li deposits without the shield was thick and comprised of abundant crystals (e.g., Li_2O) due to the Li corrosion and net growth of SEI (Supplementary Fig. S20)”. (Page 7, Line 2–4)

“Of Note, $\text{MgF}_2@\text{PVDF}$ -Cu electrode delivered the highest average CE of 96.2% for over 500 cycles, which is comparable to the results in the literature on artificial SEI or protective layers (Supplementary Table S2)”. (Page 7, Line 17–20)

“Such improvement of CE potentially relates to the uniform Li deposition and reduced corrosion guided by Li-Mg alloy and LiF at the interface⁴⁷⁻⁵⁰”. (Page 7, Line 22, 23)

“As shown in Fig. 4c, messy Li dendrites were observed on bare Cu foil (Fig. 4c). In sharp contrast, uniform Li deposits were plated on $\text{MgF}_2@\text{PVDF}$ -Cu (Fig. 4d, and Supplementary Figs. S24–S26)”. (Page 7, Line 23–26)

“EELS was conducted by a FEI Titan G2 transmission electron microscope with an

aberration corrector for a condenser lens, operated at an accelerating voltage of 300 kV". (Page 10, Line 22, 23)

Fig. R11 | Li deposits underneath the MgF₂@PVDF shield. **a** Cross-sectional SEM image of Li deposits on MgF₂@PVDF shield coated Cu foil. Inset is the digital photo of MgF₂@PVDF shield coated Cu foil, and the shield was peeled off to show the Li deposits. **b, c** Top-view SEM images of Li deposits underneath the MgF₂@PVDF shield. The electrolyte used was 1.0 M LiTFSI-DOL/DME with 1.0 wt% LiNO₃.

Fig. R12 | Li deposits on bare Cu foil and underneath the MgF₂@PVDF shield. **a** SEM image of Li deposits on bare Cu foil. **b** SEM image of Li deposits on MgF₂@PVDF shield coated Cu foil. **c** SEM image of MgF₂@PVDF shield. **d** SEM image of Li deposits underneath the MgF₂@PVDF shield. The Li deposition capacity was 3.0 mAh cm⁻² and the electrolyte used was 1.0 M LiTFSI-DOL/DME with 1.0 wt% LiNO₃.

Fig. R13 | Morphological, structural and chemical information of $\text{MgF}_2\text{@PVDF-Li}$. **a** SEM image of $\text{MgF}_2\text{@PVDF-Li}$. **b** Cryo-TEM image of $\text{MgF}_2\text{@PVDF-Li}$. **c, d** HR-TEM images of $\text{MgF}_2\text{@PVDF-Li}$. **e** EELS maps of $\text{MgF}_2\text{@PVDF-Li}$. The electrolyte used was 1.0 M LiTFSI-DOL/DME with 1.0 wt% LiNO_3 .

Fig. R14 | Cryo-TEM images of Li deposits on bare Cu grid. **a** Cryo-TEM image of corroded Li. **b** HR-TEM images of corroded Li with abundant crystals like Li_2O . The Li deposits were formed in 1.0 M LiTFSI-DOL/DME with 1.0 wt% LiNO_3 .

Comment 12: SI, Figure S1. Please specify all experimental conditions (electrochemical system, current density, temperature etc.).

Response:

Thank the reviewer for this reminding. The detailed experimental conditions have been supplemented in the revised supplementary information, which were also presented below.

“Supplementary Fig. S3 | Electrochemical protocols for detecting Li corrosion and SEI dissolution. a Voltage profiles of Li | Cu cells with different rest of time for detecting Li corrosion. The cell was cycled at 1.0 mA cm^{-2} and room temperature with a Li cycling capacity of 1.0 mAh cm^{-2} . b Voltage profiles of Li | Cu cells in the range from 0.05 V to 2.0 V with different rest of time for detecting SEI dissolution. The cell was cycled at $6.0 \mu\text{A}$ and room temperature, and the electrolyte used was 1.0 M LiTFSI-DOL/DME with 1.0 wt% LiNO_3 . c SEI dissolution related capacity loss of active Li sources in various electrolytes as a function of rest time. The electrolytes include 1.0 M LiTFSI in DOL/DME (1:1 by volume), LiTFSI in DOL/DME with 1.0 wt% LiNO_3 (1:1 by volume), LiPF_6 in EC/DMC/DEC (1:1:1 by volume), LiPF_6 in EC/EMC/DEC with 1.0 wt% FEC (1:1:1 by volume), and LiPF_6 in FEC/DME (1:4 by volume). d A schematic illustration of the SEI progression, and its correlation with the net growth of SEI and continuous Li corrosion”. (Supplementary information, Page S4, Line 1–12)

Comment 13: SI, Line 20: “Cell potential” is not correct term. Strictly speaking, cell voltage (correct term) is a difference of electrode potentials.

Response:

We’re sorry for such mistakes. We have carefully checked the whole manuscript to correct such terms, which were also presented below.

“The slight increase was observed in the cell voltage during the pause, which suggested that the SEI was not sufficiently stable and dissolved in the electrolyte”. (Supplementary information, Page S4, Line 13, 14)

Comment 14: SI, Line 20-26: In my opinion, this statement is unsupported claim until authors will demonstrate that SEI layer was really dissolved, and Li metal remained intact.

Response:

Thank the reviewer for the valuable questions. We have supplemented convincing

evidence to validate the SEI dissolution. Firstly, we prepared SEI on the Cu foil, and compared the morphological, structural and chemical changes of SEI before and after soaking in the electrolyte. As shown in Figs. R4, R15 and R16, SEI was dissolved into the electrolyte, causing the great changes in the morphology. Notably, the SEI after soaking suffered obvious shrinkage. The elemental mappings and HRTEM further validated that the SEI after soaking comprised of abundant inorganic Li salts, indicating the greater dissolution behavior of polymeric composition in the electrolyte. Moreover, we have collected the dissolved SEI composition from the electrolyte in Fig. R17, which was expectedly proved to be comprised of abundant polymeric substances and some crystalline Li salts. These results strongly validated the dissolution behavior of SEI in the electrolyte. Notably, SEI experiences dynamic evolutions, including repeated breakage/dissolution and repair/reformation, which result in the continuous loss of active Li source and the net growth of SEI (Fig. R5d). Consequently, a growingly thick SEI will form on the surface of Li until the electrolyte is saturated by SEI or the contact between Li and electrolyte is eliminated. The relevant data and discussion have been added in the revised manuscript, which can also be found below.

“Notably, the SEI dissolution can be detected in both ether and ester electrolytes (Supplementary Fig. S3c)^{33,38}. Moreover, the SEI dissolution was directly validated by detecting the morphological, structural and chemical changes of SEI before and after soaking in the electrolyte. The SEI suffered obvious shrinkage after soaking in the electrolyte (Fig. 1g, h, and Supplementary Figs. S4–6). Additionally, the SEI after soaking contained abundant Li salts and limited polymeric composition, indicating the higher solubility of organic components (Fig. 1i and Supplementary Fig. S7). To confirm such hypothesis, the dissolved SEI compositions in the DOL/DME were collected and studied, which was found to be comprised of a large number of amorphous polymeric substances and a few crystalline Li salts (Fig. 1j)”. (Page 4, Line 21–31)

Fig. R15 | Cryo-TEM images of SEI residual on bare Cu grid. a–g Cryo-TEM image, HAADF image, and elemental mappings of SEI residual on bare Cu grid. The SEI residual was formed in 1.0 M LiTFSI-DOL/DME with 1.0 wt% LiNO₃ after one Li plating/stripping process, which was then washed with DOL to dissolve the residual LiTFSI salt.

Fig. R16 | Cryo-TEM images of SEI residual on bare Cu grid after soaking in the electrolyte. a, b Cryo-TEM image and HRTEM image of SEI residual after soaking in

the electrolyte. **c** The mass contents of various elements in SEI before and after soaking in the electrolyte. **b–g** HAADF image and elemental mappings of SEI residual after soaking in the electrolyte. The SEI residual was formed in 1.0 M LiTFSI-DOL/DME with 1.0 wt% LiNO₃ after one Li plating/stripping process. The formed SEI was soaked in 1.0 M LiTFSI-DOL/DME with 1.0 wt% LiNO₃.

Fig. R17 | Cryo-TEM images of precipitation from dissolved SEI. **a** A schematic illustration showing the preparation of sample by drying the solution containing dissolved SEI. **b, c** TEM and HRTEM images of precipitation after drying the solution containing dissolved SEI. **d–i** HAADF image and elemental mappings of the precipitation from dissolved SEI. The precipitation from dissolved SEI was obtained by dissolving SEI in the mixed solvent of DOL/DME, after which the transparent solution containing dissolved SEI was transferred onto the Cu grid and dried.

Reviewer #2 (Remarks to the Author):

The authors devise a M(x)F(y)@PVDF coating to reduce "corrosion" on lithium metal anodes and find improved cycling capability. Cryo-TEM and other techniques are used to characterize the SEI layers. I have major reservations about this paper and recommend a very major revision before it is reconsidered for publication.

Response:

The authors would like to first appreciate the reviewer for raising the professional comments, which helped us to improve the quality and clearly conclude the novelty of this work. After carefully addressing all the comments raised by the reviewers, we believe that the revised manuscript could be considered to be published in Nature Communications.

Comment 1: The word "corrosion" is used numerous times, yet it is unclear what precisely that means. Corrosion is a very well studied subject, and numerous categories of corrosion has been determined. See, e.g. <https://waverleybrownall.co.uk/blog/5-common-types-corrosion/>

So when the authors say "corrosion," which category is it? What is the overpotential involved? Corrosion usually occurs at an overpotential. If the authors only mean "lithium has reacted/degraded" without focusing on corrosion mechanisms, they should consider just using "aging" or "self-discharge."

Response:

Thank you for your profession comment. The Li corrosion during battery storage and operation is thought to be a dynamic and complex process, during which both chemical corrosion (*i.e.*, pitting corrosion) and electrochemical corrosion (*i.e.*, galvanic corrosion) undergo. The native SEI suffers repeated dissolution/breakage and reformation/repair, leaving Li being exposed to electrolyte to trigger chemical corrosion. Additionally, the electrically connected Cu and Li that are exposed to the electrolyte will undergo galvanic corrosion. Li corrosion is driven by the electrochemical potential difference between the metal and the environment¹¹. **The corrosion mentioned in this work refers to the chemical corrosion and galvanic corrosion, which repeatedly undergo during battery operation and storage.** The distinguishment of corrosion patterns is

not the focus of this work. To make a better understanding of this work, we have provided more discussion on the definitions of Li corrosion in the revised manuscript, which can also be seen below.

“Consequently, the depletion of both electrolyte and active Li source constantly undergoes, triggering the localized chemical corrosion (i.e., pitting corrosion)²²”. (Page 2, Line 18–20)

“Moreover, Li corrosion can be electrochemically aggravated by a galvanic process²³, which involves a galvanic couple comprising of dissimilar metals in electrical contact (e.g., active Li and noble copper) and electrolyte. The chemical and electrochemical corrosion patterns continuously undergo during battery storage and operation, together contributing to serious Li loss²⁴”. (Page 2, Line 20–24)

Comment 2: Which electrolyte is used in Fig. 1a-d? Several papers, e.g., Adv. Energy Mater. 10, 2000017, (2020), and ACS Appl. Energy Mater. 4, 7589 (2021), have shown that the amount of "corrosion" or self-discharge depends on amount of Li deposited. How do those papers compare with the authors' results (Fig. 1f, Fig. 3a)?

Response:

Thanks for such valuable questions. The electrolytes used in Fig. 1a–d are 1.0 M LiPF₆-EC/EMC/DEC (1/1/1 by volume) with 1.0 wt% FEC and 1.0 M LiTFSI-DOL/DME (1/1 by volume) with 1.0 wt% LiNO₃. The missing information has been added in the revised manuscript, which was also presented below.

“Fig. 1 | Li corrosion and SEI dissolution. a–d Cryo-TEM images and elemental mappings of Li deposits in typical ester and ether electrolyte, respectively. The ester and ether electrolytes refer to 1.0 M LiPF₆ in EC/EMC/DEC with 1.0 wt% FEC (1:1:1 by volume) and 1.0 M LiTFSI in DOL/DME with 1.0 wt% LiNO₃, respectively. e A schematic illustration of the chemical and galvanic corrosion of Li metal. f Corrosion related capacity loss of Li and the corresponding CE as a function of rest time. g, h Cryo-TME images of SEI before and after soaking in the electrolyte. i HRTEM of SEI after soaking in the electrolyte. The electrolyte was 1.0 M LiTFSI in DOL/DME with 1.0 wt% LiNO₃. j HRTEM images of precipitation after drying the DOL/DME solvent containing dissolved SEI.”. (Page 17, Line 1–10)

In addition, we have carefully checked the two papers mentioned by the reviewers. Prof. Martin Winter, and Dr. Marian Cristian Stan systematically studied the galvanic corrosion behavior of Li powder electrodes in 1.0 M LiPF₆ in EC/DEC (3/7 by weight), where zero resistance ammetry technique was used to quantify the extent of corrosion. The initial corrosion current reaches values >160.0 $\mu\text{A cm}^{-2}$ with a subsequent decrease to values <1.0 $\mu\text{A cm}^{-2}$ already after 10.0 h (Adv. Energy Mater. 2020, 10, 2000017). Of note, the practical capacity of Li powder electrode was reduced by 0.32 mAh cm⁻² after one week aging at room temperature under open circuit voltage conditions. Prof. Katharine L. Harrison and colleagues investigated the self-discharge in high-performance Li metal battery electrolytes (4.0 M LiFSI in DME, 2.0 M LiFSI-1.0 M LiTFSI in DOL/DME) and several mitigation strategies (forming a more form a more organic-type SEI, increasing the capacity of Li plated, and lithiophilic coating). The average CEs reach 95.0–98.0% at 0.5 mA cm⁻² for 0.5 mAh cm⁻², which changed obviously with the varied rest step, current and capacity. Notably, it's found that the average CE with rest step was greater than 100%, indicating the possible capacity recovery. In terms of our work, the quantitative results of the Li corrosion triggered capacity (10.0–200.0 μAh) and CE loss (1.0–20.0%) with and without a MgF₂@PVDF shield as a function of resting time (Fig. 1f and 3a) are comparable to the findings in other papers. The papers have been cited in the revised manuscript and supplementary information, and the comparison between the literature and this work was summarized (Table R1). **In conclusion, the literature provided the quantitative recognition on Li corrosion, of which the progress promotes and inspires us to uncover the origins and solutions to Li corrosion.** In this work, the Li corrosion and SEI dissolution have been clearly validated (Fig. R18a–d). Moreover, we quantify the Li corrosion and SEI dissolution, and reveal the correlation between such two processes. **The SEI dissolution during battery storage and operation was validated to expose Li to the corrosion by the electrolyte.**

“Very recently, Cui and Zhang et al. have validated the SEI dissolution in alkali metal battery systems, which has a significant effect on the cyclability of batteries^{33,34}. However, the direct correlation between dynamic Li corrosion behaviors and SEI progression (e.g., SEI dissolution and reformation) still needs more expeditions (Supplementary Table S1)”. (Page 3, Line 3–7)

“5 Merrill, L. C., Rosenberg, S. G., Jungjohann, K. L. & Harrison, K. L. Uncovering the relationship between aging and cycling on lithium metal battery self-discharge. *ACS Appl. Energy Mater.* **4**, 7589–7598, doi:10.1021/acsaem.1c00874 (2021)”. (Supplementary Page S37, Line 10–12)

“22 Kolesnikov, A. et al. Galvanic corrosion of lithium-powder-based electrodes. *Adv. Energy Mater.* **10**, 2000017, doi:10.1002/aenm.202000017 (2020)”. (Supplementary Page S38, Line 15–17)

Fig. R18 | Li and SEI before and after soaking in the electrolyte. a, b Cryo-TEM images of Li deposits before and after soaking in the 1.0 M LiTFSI-DOL/DME with 1.0 wt% LiNO₃. **c, d** Cryo-TEM images of SEI before and after soaking in the 1.0 M LiTFSI-DOL/DME with 1.0 wt% LiNO₃.

Comment 3: The abstract and conclusion stress that dissolution of the SEI is elucidated as a main cause of "corrosion." The Introduction even claim that the manuscript has proven this point. But there is absolutely no characterization showing that the SEI has

dissolved. For example, detection of SEI fragments in the electrolyte would constitute "dissolution." Voltage-time curves (Fig. S1) do not. It is indisputable that in this aspect, the data does not support the authors' conclusion. Does Mg(2+) dissolve into the electrolyte as well?

Response:

Thank you for the profession comment. We have added direct evidence to validate the SEI dissolution. Firstly, we prepared SEI on the Cu foil/grid, and compared the morphological, structural and chemical changes of SEI before and after soaking in the electrolyte. As shown in Fig. R19a, b, the initial SEI was dense and aggregated, while the SEI after soaking in the electrolyte showed loose and dispersed morphology, indicating the dissolution of SEI. The cryo-TEM images also clearly validated that SEI was dissolved into the electrolyte and caused the obvious shrinkage of SEI (Fig. R19c, d). HRTEM image and the elemental mappings further proved that the SEI after soaking was comprised of abundant inorganic Li salts, indicating the greater dissolution behavior of polymeric composition in the electrolyte (Fig. R19e, f). Moreover, the transparent DOL/DME solution after SEI soaking was dropped onto TEM grid and dried to collect the dissolved SEI composition. It's found that the dissolved SEI composition was comprised of abundant polymeric substances and some crystalline Li salts (Fig. R19g–i). These results strongly validated the dissolution behavior of SEI in the electrolyte, where organic composition showed greater solubility. The relevant discussion and data have been added in the revised manuscript, which can also be observed below.

“Notably, the SEI dissolution can be detected in both ether and ester electrolytes (Supplementary Fig. S3c)^{33,38}. Moreover, the SEI dissolution was directly validated by detecting the morphological, structural and chemical changes of SEI before and after soaking in the electrolyte. The SEI suffered obvious shrinkage after soaking in the electrolyte (Fig. 1g, h, and Supplementary Figs. S4–6). Additionally, the SEI after soaking contained abundant Li salts and limited polymeric composition, indicating the higher solubility of organic components (Fig. 1i and Supplementary Fig. S7). To confirm such hypothesis, the dissolved SEI compositions in the DOL/DME were collected and studied, which was found to be comprised of a large number of amorphous polymeric substances and a few crystalline Li salts (Fig. 1j)”. (Page 4, Line 21–31)

Fig. R19 | Li and SEI before and after soaking. **a–d** SEM and cryo-TEM images of SEI before and after soaking in the electrolyte. **e** HRTEM of SEI after soaking in the electrolyte. **f** Elemental contents in SEI before and after soaking in the electrolyte. The electrolyte used was 1.0 M LiTFSI-DOL/DME with 1.0 wt% LiNO₃. **g** A schematic illustration showing the preparation of sample by drying the DOL/DME solution containing dissolved SEI. **h, i** TEM and HRTEM images of precipitation after drying the DOL/DME containing dissolved SEI.

In terms of the solubility of Mg²⁺ in the electrolyte, we have supplemented the quantitative elemental information from inductively coupled plasma-mass spectrometry (ICP-MS, Agilent 7850). Of note, we have tested the concentration of Mg²⁺ in solution containing MgF₂ powder or MgF₂@PVDF shield. As shown in Fig. R20, the Mg²⁺ from pristine MgF₂ powder was 0.16 mg mL⁻¹, while the Mg²⁺ from MgF₂@PVDF was decided to be 0.022 mg mL⁻¹. Obviously, limited Mg²⁺ can dissolve

into the electrolyte. Relevant figures and discussion have been also added in the revised manuscript and supplementary information.

“*MgF₂ as a typical low-solubility and lithiophilic fluoride was selected as a prototype during the investigation of the shield (Supplementary Fig. S13)⁴²*”. (Page 5, Line 17–19)

Fig. R20 | Solubility of MgF₂. a, b Digital photos of MgF₂@PVDF and MgF₂ powder soaked in the DOL/DME. c The concentration of Mg²⁺ in the solution containing MgF₂@PVDF or MgF₂ powder.

Comment 4: Without having "proven" their key hypothesis, it is unclear what the main point of the paper is, apart from the fact that the authors have a new coating that improve cycling (Fig. 5d, Fig. 5g). They cite 96.2% capacity after 500 cycles. What is the coulomb efficiency (CE) ? How does it compare with the numerous papers in the literature which use coatings to improve the CE? I guess the authors need to show the novelty in this paper and tell the reader why it is distinct from so many related papers in the literature.

Response:

We sincerely thank the reviewer for this enlightening comment. Following the reviewer’s helpful suggestion, we have concluded the main point of paper as to reveal the SEI dissolution behavior and uncover its effect on the continuous Li corrosion. The SEI dissolution behaviors have been validated and studied by supplemented cryo-TEM and SEM images in Fig. R19. In this work, a prototypical shielding strategy is proposed,

which indicates the significance of inhibiting SEI dissolution to eliminate Li corrosion (Fig. R21).

Fig. R21 | Li corrosion with and without a shield. a, b SEM images of Li deposits with and without a shield. **c** CE and capacity loss of Li | Cu cells with and without a shield. The electrolyte used was 1.0 M LiTFSI-DOL/DME with 1.0 wt% LiNO₃.

The value of 96.2% refers to the average CE of a Li | Cu cell after 500 cycles under a moderate condition (1.0 mA cm⁻², 1.0 mAh cm⁻²), where a MgF₂@PVDF shield was used. The comparison between this work and literature can be seen in Table R3. The CE enabled by the shielding strategy is comparable to the state-of-the-art performances in the literature, considering the high CE and prolonged lifespan.

Table R3. Comparison between the literature on artificial SEI/protective layer and this work.

Material	Mechanism	Condition	CE	Ref
Interconnected hollow carbon nanospheres	Dendrite suppression and interfacial stabilization	1.0/1.0	~99% for 150 cycles	20
3D oxidized polyacrylonitrile nanofiber network	Guiding the Li ⁺ to form uniform Li deposits	3.0/1.0	97.4 for 120 cycles	21
Dihydroxyviolanthron layer	Stabilizing SEI and homogenizing the deposition of Li metal	0.5/1.0	99.6 for 300 cycles	22
Lithiated Nafion film reinforced by organic filler	Producing a flexible interphase with uniform Li ⁺ diffusion	1.0/1.0	99.2% for ~150 cycles	23

Li ₂ S/Li ₂ Se protective layer	Constructing stable surface SEI with high ionic conductivity	1.0/1.0	98% for 360 cycles	24
Dual-protective interface of Prussian blue/rGO	Facilitating fast and uniform Li ⁺ flux to achieve uniform Li deposition	1.0/1.0	97.8% for 150 cycles	25
Poly(dimethylsiloxane) film with nanopores	Restraining Li dendrite growth and reducing the side reaction	1.0/1.0	93.2% for 100 cycles	26
A garnet LLZTO/lithiated Nafion layer	Affording uniform distribution/transport of Li ⁺ , and accommodating Li deposition	1.0/1.0	97.7% for 150 cycles	27
A polymeric ionic liquid	Unifying Li ⁺ flux and promoting a homogeneous Li plating	0.5/1.0	96.9% for 110 cycles	28
Cu ₃ N/styrene butadiene rubber	Facilitating the transport of Li ⁺ and maintaining structural integrity	1.0/1.0	97.4% for 100 cycles	29
Supramolecular copolymer layer	Affording a high self-stabilization and strong adhesion	1.0/1.0	98.42% for 150 cycles	30
Poly(vinylidene-co-hexafluoropropylene)/LiF	Suppressing random Li deposition and the formation of isolated Li	1.0/1.0	96.3% for 60 cycles	31
Dynamic single-ion-	Mitigating side	1.0/1.0	94.9%	32

conductive network	reactions and regulating Li deposition		for 250 cycles	
Langmuir–Blodgett artificial SEIs	Inhibiting parasitic reactions and enabling uniform Li electrodeposition.	1.0/1.0	~96% for ~200 cycles	33
A reactive polymer composite derived SEI and 3D host	stabilizing the interface and preventing electrolyte decomposition	2.0/4.0	99.1% for 300 cycles	34
A salt-philic, solvent-phobic polymer coating	Promoting salt-derived SEI formation	0.5/1.0	99.5% for 10 cycles	35
A shielding layer of polymer and fluorides	Inhibiting SEI dissolution and Li corrosion	1.0/1.0	96.2% for 500 cycles	This work

Condition refers to the current density (mA cm^{-2}) and cycling capacity (mAh cm^{-2}) corresponding to the recorded CE in the table.

The Li corrosion and dynamic SEI evolution (*e.g.*, swelling) are emerging topics in researches of Li metal anode^{5,36}. Most of the previous researches are devoted to constructing stable interfaces to promise uniform Li^+ flux and guide the dendrite-free Li deposition by artificial SEI/protective layer (Table R3). However, the influence of such layers on the suppression of Li corrosion and SEI dissolution remains to be further studied. Our work aimed to reveal the SEI dissolution behaviors and uncovered its correlation with Li corrosion. The shielding strategy by a composite coating layer is a prototype to inhibit SEI dissolution and Li corrosion, which can be further extended (*e.g.*, tuning the electrolyte solvation, isolating the contact between Cu and electrolyte, recovering the corroded Li and dissolved SEI). Relevant discussion has been added in the revised manuscript, which can also be seen below.

“The role of artificial SEI has been extendedly understood from the perspective of anti-corrosion, which distinguishes from the conventional designs”. (Page 3, Line 23, 24)

“Of note, PVDF-Cu had an improved CE of 94.2% at the 297th cycle. MgF₂@PVDF-Cu electrode delivered the highest average CE of 96.2% for over 500 cycles, which is comparable to the results in the literature on artificial SEI or protective layers (Table S2)”. (Page 7, Line 17–20)

Reviewer #3 (Remarks to the Author):

The authors report how a fluoride/PVDF coating can help suppress corrosion of Li. The report that coatings can help prevent corrosion of Li is notable and remains an area to explore further. Especially as a method to augment electrolyte engineering. This alone is an interesting topic, but several points are necessary to address below to make this paper of suitable quality in this reviewer's opinion.

Response:

We thank the reviewer for the professional and valuable comments, which greatly help us to improve the quality of this work. Great progress has been made in stabilizing Li metal by electrolyte and interfacial engineering. Very recently, SEI evolution and Li corrosion have been taken into account inspired by such previous researches. Li corrosion mechanism and inhibiting strategy should be concerned and explored. On basis of the suggestions from the reviewer, we have carefully revised the manuscript with necessary data to confirm the findings and conclusions in this work.

Comment 1: Certain portions of the writing are over exaggerated: This work unveils the origin of Li corrosion" and "the direct correlation between dynamic Li corrosion behaviors and SEI progression presently remains unknown" - the connection of SEI progression with corrosion was investigated closely in reference 30, as well as others - doi: 10.1021/jacs.0c10258, 10.1002/aenm.202202012,

Response:

We highly appreciate the reviewer for commending on the writing of this work, and have carefully corrected such descriptions in the whole text. Besides, we agree that the connection of SEI progression with corrosion has been investigated in several papers. However, **it should be noted that those papers attributed the growth of SEI to Li corrosion, while our work aimed to reveal the influence of SEI dissolution on continuous Li corrosion.** For instance, Cui *et al.* quantified the impact of calendar ageing on the rechargeability of Li metal anodes⁵. It's found that continuous growth of the SEI via chemical corrosion causes these losses of capacity. Grey *et al.* developed in situ NMR metrology to quantify the dead Li and SEI formation, demonstrating Li corrosion to be a critical issue. The high rate of corrosion is attributed to SEI formation on both Li metal and copper. In the study of Meng *et al.*, a quantitative relationship

between the chemical corrosion rate and electro-chemically deposited Li morphology (e.g., porosity) is established¹¹. Additionally, **we have directly validated the SEI dissolution and Li corrosion with cryo-TEM and electrochemical protocols.** Those papers were cited and discussed in the revised manuscript, which were also presented below.

“This work uncovers the SEI dissolution and its correlation with Li corrosion, enabling the durable operation of Li metal batteries by interdicting the Li loss”. (Page 1, Line 24–26)

“Very recently, Cui and Zhang et al. have validated the SEI dissolution in alkali metal battery systems, which has a significant effect on the cyclability of batteries^{33,34}. However, the correlation between dynamic Li corrosion behaviors and SEI progression (e.g., SEI dissolution and reformation) still needs more expeditions (Supplementary Table S1)”. (Page 3, Line 3–7)

“Most importantly, this work uncovers the dynamic Li corrosion and SEI dissolution, which promises Li metal batteries with ultralong lifespans by reducing the corrosion-induced Li loss.” (Page 3, Line 25–27)

“24 Lu, B. et al. Suppressing chemical corrosions of lithium metal anodes. Adv. Energy Mater. 12, 2202012, doi:10.1002/aenm.202202012 (2022)”. (Page 13, Line 31, 32)

“31 Gunnarsdottir, A. B., Amanchukwu, C. V., Menkin, S. & Grey, C. P. Noninvasive in situ NMR study of "dead lithium" formation and lithium corrosion in full-cell lithium metal batteries. J. Am. Chem. Soc. 142, 20814-20827 (2020)”. (Page 14, Line 15–18)

Comment 2: Certain areas need more support or explanation: "Notably, the SEI dissolution can be detected in various ether and ester electrolytes" - is there a reference or measurement to support this claim? dissolution of the SEI is proposed to be a major cause of corrosion and a key finding in the abstract (mentioned twice), but it isn't tracked at all in the manuscript.

Response:

We thank the reviewer for suggesting us to provide more support and explanation. To validate the descriptions in our work, we firstly have added references to support the

SEI dissolution in LiFSI-DME/ 1,1,2,2-tetrafluoroethyl 2,2,3,3-tetrafluoropropyl ether², LiTFSI-LiNO₃-DOL/DME³, LiPF₆-EC/DEC⁵, LiPF₆-EC/DMC³⁷, etc. We have also conducted several experiments to validate the SEI dissolution. We prepared SEI on the Cu foil/grid, and compared the morphological, structural and chemical changes of SEI before and after soaking in the electrolyte. As shown in Fig. R22a, b, the initial SEI was dense and aggregated, while the SEI after soaking in the electrolyte showed loose and dispersed morphology, indicating the dissolution of SEI. The cryo-TEM images also clearly validated that SEI was dissolved into the electrolyte and caused the obvious shrinkage of SEI (Fig. R22c, d). HRTEM image further proved that the SEI after soaking was comprised of abundant inorganic Li salts, indicating the greater dissolution behavior of polymeric composition in the electrolyte (Fig. R22e). Moreover, the transparent DOL/DME solution after SEI soaking was dropped onto TEM grid and dried to collect the dissolved SEI composition. It's found that the dissolved SEI composition was comprised of abundant polymeric substances and some crystalline Li salts (Fig. R22f-h). Besides, our electrochemical protocols also validated the SEI dissolution in various electrolytes (Fig. R22i). These results strongly validated the dissolution behavior of SEI in the electrolyte, where organic composition showed greater solubility. Such SEI dissolution will expose Li to the electrolyte, triggering continuous corrosion.

“Notably, the SEI dissolution can be detected in both ether and ester electrolytes (Supplementary Fig. S3c)^{33,38}. Moreover, the SEI dissolution was directly validated by detecting the morphological, structural and chemical changes of SEI before and after soaking in the electrolyte. The SEI suffered obvious shrinkage after soaking in the electrolyte (Fig. 1g, h, and Supplementary Figs. S4–6). Additionally, the SEI after soaking contained abundant Li salts and limited polymeric composition, indicating the higher solubility of organic components (Fig. 1i and Supplementary Fig. S7). To confirm such hypothesis, the dissolved SEI compositions in the DOL/DME were collected and studied, which was found to be comprised of a large number of amorphous polymeric substances and a few crystalline Li salts (Fig. 1j)”. (Page 4, Line 21–31)

Fig. R22 | SEI dissolution. **a–d** SEM and cryo-TEM images of SEI before and after soaking in the electrolyte. **e** HRTEM image of SEI after soaking in the electrolyte. The electrolyte used was 1.0 M LiTFSI-DOL/DME with 1.0 wt% LiNO₃. **f** A schematic illustration showing the preparation of sample by drying the DOL/DME solution containing dissolved SEI. **g, h** TEM and HRTEM images of precipitation after drying the solution containing dissolved SEI. **i** Capacity loss triggered by SEI dissolution in various electrolytes.

Comment 3: Fluorides have low ionic conductivity. Why are the authors referring to a fluoride based SEI as having high conductivity? Furthermore, although the literature commonly claims high ionic conductivity of the SEI is needed for better cyclability, recent authoritative work that drew correlations across many electrolytes have suggested this is not a leading predictor of high CE: doi: 10.1038/s41560-021-00910-w (Nature Energy), 10.1021/jacs.2c08182 (JACS), 10.1038/s41560-022-01144-0

(Nature Energy)

Response:

We agree with the reviewer that fluorides have low ionic conductivity. Such description that fluoride based SEI has high conductivity has been carefully checked and corrected in the whole text. Besides, the reason for selecting fluorides as the inorganic composition of a shield has been concluded. Firstly, the fluoridation and lithiophilic strategy has been systematically studied in our previous works to construct a stable LiF-rich SEI and regulate the uniform Li deposition^{13,16,17}. Notably, the enriched LiF shows stability at the Li^+/Li^0 potential, negligible solubility in electrolyte and poor electrical conductivity³, which can effectively isolate the electrolyte penetration and corrosion. The selected fluorides can react with Li to form LiF and Li alloys at the interface. Such a stabilized SEI contributes to uniform Li^+ transport, dendrite-free Li deposition and protected Li³⁸.

Additionally, we agree that high conductivity is not a leading predictor of high CE. The improved CE in this work can be ascribed to the introduction of the shield, which promises a stable interface and inhibits the Li corrosion. We have provided more discussion on the improved CE in the revised manuscript and included the mentioned papers in the references, which can also be seen below.

“*Such improvement of CE potentially relates to the uniform Li deposition and reduced corrosion guided by Li-Mg alloy and LiF at the interface⁴⁷⁻⁵⁰*”. (Page 7, Line 22, 23)

47 Kong, L. L. et al. Lithium–magnesium alloy as a stable anode for lithium–sulfur battery. *Adv. Funct. Mater.* 29, 1808756 (2019).

48 Hobold, G. M. et al. Moving beyond 99.9% Coulombic efficiency for lithium anodes in liquid electrolytes. *Nat. Energy* 6, 951–960 (2021).

49 Ko, S. et al. Electrode potential influences the reversibility of lithium-metal anodes. *Nat. Energy* 7, 1217–1224 (2022).

50 Boyle, D. T. et al. Correlating kinetics to cyclability reveals thermodynamic origin of lithium anode morphology in liquid electrolytes. *J. Am. Chem. Soc.* 144, 20717–20725 (2022)”. (Page 15, Line 24–32)

Comment 4: The SEM and TEM images of Li and the polymer coating are not aligned.

Why in the SEM images is the polymer thickness 25 microns thick, but in TEM the polymer thickness is believed to be only 10 nm thick. EELS is the better method to assign chemistry especially when many materials appear amorphous or are not aligned clearly to lattice fringes.

Response:

We thank the reviewer for such professional comments. The shielding layers can be facily prepared on the surface of Li foil or current collectors by a blade coating strategy, and the thickness can be well controlled. The SEM images showed the morphology and thickness of the shield on the Li foils, which were later used in the electrochemical evaluation. In terms of the shield for cryo-TEM test, the thickness was purposely reduced with the aim to visualize the microstructure of crystalline substances in the polymer matrix. An ultra-large thickness of polymer will prevent the acquirement of high-resolution TEM images (Fig. R23). We have explained the reason for reducing the thickness of polymer in the cryo-TEM study in the revised manuscript, which can also be found below.

“Notably, dilute solution of PVDF and fluoride was applied to reduce the thickness of shield with the purpose of better visualizing the interface during TEM characterization”.

(Page 10, Line 20, 21)

Fig. R23 | Cryo-TME image of Li deposits with a thick MgF₂@PVDF shield.

Besides, on basis of the reviewer’s suggestion, we have supplemented the EELS maps

of Li depositing underneath the MgF₂@PVDF. As shown in Fig. R24, elemental F and Mg of the shield mainly exist on the surface of deposited Li, indicating the uniform coating of the shield on Li deposits. We have provided the EELS images and relevant discussion in the revised paper, which were also presented below.

“Electron energy loss spectroscopy (EELS) mapping were further carried out to witness the elemental distributions of multi-layered interface of Li deposits (Fig. 3h), indicating the tight adhesion of shield on the surface of Li deposits”. (Page 7, Line 4–7)

Fig. R24 | EELS maps of Li deposits with a MgF₂@PVDF shield.

Comment 5: For the Tafel plot analysis. The low over potential slope, rather than linear extrapolation from high over potential regions, should be used to calculate the exchange current density. Many reports in the literature also make this mistake. This may fix the discrepancy between the EIS and Tafel data, though. Right now Fig. 3b,c are not in agreement as would be expected for such an analysis. Furthermore, why is the exchange current density increasing with time more for MgF₂/PVDF coating? Usually the resistance of the interface increases with rest time because the SEI grows.

Response:

Thank you for commenting on the Tafel plot analysis. We have recalculated the exchange current density on basis of low over potential region (–50 to 25 mV) and redraw the figure (Fig. R25a, b). As is known, the exchange current density can also be extracted from the EIS fitting³⁹. On basis of Fig. 3b, it seems that the pristine Li | Cu cell with lower resistance should deliver larger exchange current density compared to cells with a PVDF or MgF₂@PVDF shield. However, noted that EIS in Fig. 3b was

collected from fresh Li | Cu cells at open-circuit voltage, where the SEI has not been completely formed. As widely detected in the literature⁴⁰, the cell impedance without a coating layer will suffer continuous growth after being operated, inducing a much larger cell impedance. In contrast the cell impedance with a shield can stay stable (Fig. R25c, d). Moreover, the main purpose of this result was to validate that the introduction of shield will not largely increase the cell resistance. Conclusively, the mentioned discrepancy between the EIS and Tafel data was thought to be triggered by the cell configuration, cell status and electrochemical protocols.

Fig. R25 | Exchange current and EIS of cells. **a, b** Tafel plots and corresponding exchange current densities obtained from Li | Li cells with bare Li foil, PVDF-Li, and MgF₂@PVDF-Li, respectively. **c, d** EIS of Li | Cu cells with bare Li foil, PVDF-Li, and MgF₂@PVDF-Li before cycling and after cycling, respectively. The electrolyte used was 1.0 M LiTFSI-DOL/DME with 1.0 wt% LiNO₃.

We agree that the resistance of the interface increases with the rest time. The exchange

current density reflects the intrinsic kinetics of electron transfer coupled to solvation and ion transfer of Li^+ to form Li^0 , while the existence of an SEI largely bottlenecks such processes. As to the increase of exchange current density as a function of cycle number, it may be ascribed to the properties and evolutions of SEI during cycling. Recently, Gallant *et al.* has also validated the increase of exchange current density as a function of CV scan number in DOL/DME based ether electrolyte³⁹. Such increases in exchange current density over cycling may have two origins: (1) an increase in the intrinsic rate of Li^+ exchange of the SEI (*e.g.*, chemical composition changes over cycling); (2) an increase in the electrochemically-active area for Li^+/Li^0 redox due to surface roughening³⁹. With the adoption of $\text{MgF}_2@\text{PVDF}$ shield, the native SEI is stabilized and the interfacial kinetic is enhanced, consequently exhibiting a large exchange current density. Additionally, the gradual electrolyte penetration and swelling of PVDF promises the better solvation of Li^+ ions at the Li-polymer interface⁴¹. This allows for easier dissolution and solvation of the Li from the metal electrode surface and could also increase the concentration of Li^+ ions available for deposition locally. These phenomena would also cause an increase in exchange current density. The mentioned changes and discussion have been added in the revised manuscript, which were also presented below.

“The gradual increase of exchange current density may be ascribed to the increase in the intrinsic rate of Li^+ exchange of the SEI or the electrochemically-active area for Li^+/Li^0 redox^{40,46}”. (Page 6, Line 23–26)

“The tafel plot of the log current versus cell polarization was linearly fit over the voltage range from -25 to -50 mV, after which exchange current density can be calculated”. (Page 11, Line 16–18)

*“46 Hobold, G. M., Kim, K.-H. & Gallant, B. M. Beneficial vs. inhibiting passivation by the native lithium solid electrolyte interphase revealed by electrochemical Li^+ exchange. *Energy & Environ. Sci.* **16**, 2247–2261 (2023).”* (Page 15, Line 21–23)

Comment 6: For the battery data. I don't believe the loading is reported for the coin cell data in Fig. 5 a-c, making the C-rates and capacity retention in the coin cells meaningless at this stage. It's notable that while 10C and 1500 cycles are advertised here, 0.2 C is used for the pouch cell that only reached 70 cyc with a high N/P of 4.3.

As mentioned in the beginning of this review, the clear demonstration that such coatings can prevent corrosion is itself valuable, but exaggeration of battery cyclability isn't helpful.

Response:

We appreciate for the kind reminding of the reviewer. We have edited the inadaptable descriptions, and provided the cell parameters and testing conditions in the figures containing electrochemical cycling tests (Fig. R26), making better evaluation of battery performances obtained by the shielding strategy. The difference in performances of coin cell and pouch cells mainly originates from the amount of Li (400 μm versus 50 μm), mass loading of cathode (1.2 mAh cm^{-2} versus 2.3 mAh cm^{-2}) and the pouch cell technology. Besides, the performance comparison between control group and experimental group either in coin cell and pouch cell was reasonable, since the cell parameters and testing conditions in each test were kept the same. Moreover, we have received the suggestion of the reviewer to provide more evidence and discussion to validate the anticorrosion of the shields. As shown in Fig. R27a, b, the Li deposits after soaking in the electrolyte suffered obvious corrosion with a large number of voids. In sharp contrast, the introduction of $\text{MgF}_2@\text{PVDF}$ shield can prevent the corrosion of Li at both low and high deposition capacity of Li (Fig. R27c–f). The mentioned changes in the revised manuscript can also be found below.

Fig. R26 | Full cell evaluation in pair with LiFePO₄ cathode. **a** Rate capability of Li | LiFePO₄ cells with and without a MgF₂@PVDF shield at 0.1, 0.2, 0.5, 1, 2, 5, 10, and 1C. **b, c** Voltage profiles of Li | LiFePO₄ cells without and with a MgF₂@PVDF shield, respectively. **d, e** Prolonged cycling performance of Li | LiFePO₄ cells with and without a MgF₂@PVDF shield at 1C, and the corresponding CE. **f** Schematic illustration of a MgF₂@PVDF-Li | LiFePO₄ cell. Insets are the cell parameters. **g** Cycling performances of Li | LiFePO₄ pouch cells with and without a MgF₂@PVDF shield.

Fig. R26 | Li deposits with and without a shield. **a, b** SEM images of Li deposits (0.5 mAh cm^{-2}) after soaking in the electrolyte without the protection of a shield. **c, d** SEM images of Li deposits (0.5 mAh cm^{-2}) after soaking in the electrolyte with the protection of a shield. **e, f** SEM images of Li deposits (3.0 mAh cm^{-2}) after soaking in the electrolyte with the protection of a shield.

“Consequently, Li | Cu and Li | LiFePO₄ cells with superiorly stable interfaces can be enabled to deliver high average Coulombic efficiency (96.2% for 500 cycles) and superior cycling stability. Such a shielding strategy can be preliminarily translated into real ampere-hour-scale Li | LiFePO₄ and Li | LiNi_{0.5}Co_{0.2}Mn_{0.3}O₂ (NCM523) pouch cells. The role of artificial SEI has been extendedly understood from the perspective of anti-corrosion, which distinguishes from the conventional designs. Most importantly, this work uncovers the dynamic Li corrosion and SEI dissolution, which promises Li metal batteries with ultralong lifespans by reducing the corrosion-induced Li loss.”.

(Page 3, Line 19–27)

“Prolonged operations of Li symmetric cells and LiFePO₄ full cells with reduced Li corrosion by 86% are achieved (0.7 versus $4.8 \mu\text{Ah h}^{-1}$).” (Page 1, Line 21–23)

“In sharp contrast, uniform Li deposits were plated on MgF₂@PVDF-Cu (Fig. 4d, and Supplementary Figs. S24–S26). Moreover, with the adoption of MgF₂@PVDF, the Li deposits can maintain the corrosion-free morphology after being soaked in the

electrolyte, proving its efficiency as an anti-corrosive shield (Supplementary Fig. S27)”.

(Page 7, Line 24–28)

“The Li deposits after soaking in the electrolyte suffered obvious corrosion with a large number of voids. In sharp contrast, the introduction of $\text{MgF}_2@\text{PVDF}$ shield can prevent the corrosion of Li at both low and high deposition capacity of Li”. (Supplementary information, Page S28, Line 7–9)

“The feasibility of $\text{MgF}_2@\text{PVDF}$ shield was preliminarily validated in real ampere-hour-scale pouch cells with high-loading LiFePO_4 (2.3 mAh cm^{-2}) and NCM523 (3.2 mAh cm^{-2}).” (Page 8, Line 33, Page 9, Line 1, 2)

“However, the practical application of such a strategy in real cells need to be further optimized and developed”. (Page 9, Line 7, 8)

“Consequently, both coin-type and pouch cells exhibited superior cycling stability and remarkably extended lifespans”. (Page 9, Line 15, 16)

References

- 1 Jin, Y. *et al.* Low-solvation electrolytes for high-voltage sodium-ion batteries. *Nat. Energy* **7**, 718–725 (2022).
- 2 Sayavong, P. *et al.* Dissolution of the solid electrolyte interphase and its effects on lithium metal anode cyclability. *J. Am. Chem. Soc.*, doi:10.1021/jacs.3c03195 (2023).
- 3 Lin, D. *et al.* Fast galvanic lithium corrosion involving a Kirkendall-type mechanism. *Nat. Chem.* **11**, 382–389 (2019).
- 4 Kolesnikov, A. *et al.* Galvanic corrosion of lithium-powder-based electrodes. *Adv. Energy Mater.* **10**, 2000017 (2020).
- 5 Boyle, D. T. *et al.* Corrosion of lithium metal anodes during calendar ageing and its microscopic origins. *Nat. Energy* **6**, 487–494 (2021).
- 6 Merrill, L. C., Rosenberg, S. G., Jungjohann, K. L. & Harrison, K. L. Uncovering the relationship between aging and cycling on lithium metal battery self-discharge. *ACS Appl. Energy Mater.* **4**, 7589–7598 (2021).
- 7 Li, J. *et al.* Strategies to anode protection in lithium metal battery: A review. *InfoMat* **3**, 1333–1363 (2021).

- 8 Kang, D., Xiao, M. & Lemmon, J. P. Artificial solid-electrolyte interphase for lithium metal batteries. *Batteries & Supercaps* **4**, 445–455 (2020).
- 9 Ma, C. *et al.* Chemically induced activity recovery of isolated lithium in anode-free lithium metal batteries. *Nano Lett.* **22**, 9268–9274 (2022).
- 10 Bi, C.-X. *et al.* Protecting lithium metal anodes in lithium–sulfur batteries: A review. *Energy Mater. Adv.* **4**, 0010 (2023).
- 11 Lu, B. *et al.* Suppressing chemical corruptions of lithium metal anodes. *Adv. Energy Mater.* **12**, 2202012 (2022).
- 12 Ding, J. F. *et al.* Dynamic galvanic corrosion of working lithium metal anode under practical conditions. *Adv. Energy Mater.*, 2204305 (2023).
- 13 Jin, C. *et al.* Metal oxide nanoparticles induced step-edge nucleation of stable Li metal anode working under an ultrahigh current density of 15 mA cm^{-2} . *Nano Energy* **45**, 203–209 (2018).
- 14 Jin, C. *et al.* 3D lithium metal embedded within lithiophilic porous matrix for stable lithium metal batteries. *Nano Energy* **37**, 177–186 (2017).
- 15 Liu, T. *et al.* In-situ construction of a Mg-modified interface to guide uniform lithium deposition for stable all-solid-state batteries. *J. Energy Chem.* **55**, 272–278 (2021).
- 16 Sheng, O. *et al.* In situ construction of a LiF-enriched interface for stable all-solid-state batteries and its origin revealed by cryo-TEM. *Adv. Mater.* **32**, 2000223 (2020).
- 17 Liu, Y. *et al.* Self-assembled monolayers direct a LiF-rich interphase toward long-life lithium metal batteries. *Science* **375**, 739–745 (2022).
- 18 Kong, L. L. *et al.* Lithium–magnesium alloy as a stable anode for lithium–sulfur battery. *Adv. Funct. Mater.* **29**, 1808756 (2019).
- 19 Choi, S. H. *et al.* Marginal magnesium doping for high-performance lithium metal batteries. *Adv. Energy Mater.* **9**, 1902278 (2019).
- 20 Zheng, G. *et al.* Interconnected hollow carbon nanospheres for stable lithium metal anodes. *Nat. Nanotechnol.* **9**, 618–623 (2014).
- 21 Liang, Z. *et al.* Polymer nanofiber-guided uniform lithium deposition for battery electrodes. *Nano Lett.* **15**, 2910–2916 (2015).
- 22 Chang, S. *et al.* In situ formation of polycyclic aromatic hydrocarbons as an artificial hybrid layer for lithium metal anodes. *Nano Lett.* **22**, 263–270 (2022).

- 23 Li, S. *et al.* A robust all-organic protective layer towards ultrahigh-rate and large-capacity Li metal anodes. *Nat. Nanotechnol.* **17**, 613–621 (2022).
- 24 Liu, F. *et al.* A mixed lithium-ion conductive Li₂S/Li₂Se protection layer for stable lithium metal anode. *Adv. Funct. Mater.* **30**, 2001607 (2020).
- 25 Fan, L. *et al.* A dual-protective artificial interface for stable lithium metal anodes. *Adv. Energy Mater.* **11**, 2102242 (2021).
- 26 Zhu, B. *et al.* Poly(dimethylsiloxane) thin film as a stable interfacial layer for high-performance lithium-metal battery anodes. *Adv. Mater.* **29**, 1603755 (2017).
- 27 Xu, R. *et al.* Dual-phase single-ion pathway interfaces for robust lithium metal in working batteries. *Adv. Mater.* **31**, 1808392 (2019).
- 28 Wu, J. *et al.* Polycationic polymer layer for air-stable and dendrite-free li metal anodes in carbonate electrolytes. *Adv. Mater.* **33**, 2007428 (2021).
- 29 Liu, Y. *et al.* An artificial solid electrolyte interphase with high li-ion conductivity, mechanical strength, and flexibility for stable lithium metal anodes. *Adv. Mater.* **29**, 1605531 (2017).
- 30 Wang, G. *et al.* Self-stabilized and strongly adhesive supramolecular polymer protective layer enables ultrahigh-rate and large-capacity lithium-metal anode. *Angew. Chem. Int. Ed.* **59**, 2055–2060 (2020).
- 31 Xu, R. *et al.* Artificial soft-rigid protective layer for dendrite-free lithium metal anode. *Adv. Funct. Mater.* **28**, 1705838 (2018).
- 32 Yu, Z. *et al.* A dynamic, electrolyte-blocking, and single-ion-conductive network for stable lithium-metal anodes. *Joule* **3**, 2761–2776 (2019).
- 33 Kim, M. S. *et al.* Langmuir–Blodgett artificial solid-electrolyte interphases for practical lithium metal batteries. *Nat. Energy* **3**, 889–898 (2018).
- 34 Gao, Y. *et al.* Polymer-inorganic solid-electrolyte interphase for stable lithium metal batteries under lean electrolyte conditions. *Nat. Mater.* **18**, 384–389 (2019).
- 35 Huang, Z. *et al.* A salt-philic, solvent-phobic interfacial coating design for lithium metal electrodes. *Nat. Energy*, doi:10.1038/s41560-023-01252-5 (2023).
- 36 Zhang, Z. *et al.* Capturing the swelling of solid-electrolyte interphase in lithium metal batteries. *Science* **37**, 66–70 (2022).
- 37 Jin, Y., Kneusels, N. H. & Grey, C. P. NMR study of the degradation products

- of ethylene carbonate in silicon-lithium ion batteries. *J. Phys. Chem. Lett.* **10**, 6345–6350 (2019).
- 38 Kim, M. H. *et al.* Design principles for fluorinated interphase evolution via conversion-type alloying processes for anticorrosive lithium metal anodes. *Nano Lett.* **23**, 3582–3591 (2023).
- 39 Hobold, G. M., Kim, K.-H. & Gallant, B. M. Beneficial vs. inhibiting passivation by the native lithium solid electrolyte interphase revealed by electrochemical Li⁺ exchange. *Energy & Environ. Sci.* **16**, 2247–2261 (2023).
- 40 Chen, H. *et al.* Uniform high ionic conducting lithium sulfide protection layer for stable lithium metal anode. *Adv. Energy Mater.* **9**, 1900858 (2019).
- 41 Lopez, J. *et al.* Effects of polymer coatings on electrodeposited lithium metal. *J. Am. Chem. Soc.* **140**, 11735–11744 (2018).

REVIEWER COMMENTS

Reviewer #1 (Remarks to the Author):

Authors significantly improved quality of the revised manuscript addressing comments mentioned in the 1st review. Therefore, I would like to recommend the manuscript for publishing in Nature Communications after addressing all points described below:

All text including title: Despite demonstration of certain SEI dissolution, I would not attribute the observed improvement only by this factor since literature indicates many other causes. I think the interface protection of Li anode should be mentioned in more broad discussion because, as it was proven before, irreversible Li loss have several causes. Moreover, conceptually speaking, Li metal will always have SEI layer in corresponding electrolytes. IMO, this simple argument clearly shows limitation of 100% focus of authors on the corrosion.

Line 203. As it was mentioned by one of the reviewers, please do not exaggerate some points. For example, "corrosion-free" which is very brave and unsupported claim.

Line 267. "full cell" appears again. Please check.

Conclusions: I think this part should be improved via providing more concrete results (for example, the best chemistry for protection layer).

Figure 5f: I think this drawing should be removed.

Reviewer #2 (Remarks to the Author):

The authors have extensively revised their manuscript and added evidence of SEI dissolution. They have also changed their abstract so it no longer claims to show proof of SEI dissolution causing Li "corrosion" (it can easily be the other way round) -- that is much appreciated. While Referee 1 helpfully pointed out several review articles documenting PVDF/MgF₂ as a coating agent, it is true, as the authors point out, that most of the work reviewed there is not related to Li "corrosion."

Overall I now recommend publishing this work provided the two points below are addressed (not peripherally, but in some detail).

One issue the authors have not raised is the self-discharge time scale involved. The authors go to ~50 hours; is this sufficient? One of the reasons silicon anodes continue to struggle to be commercialized is their limited calendar life, which is related to self-discharge rate. Researchers in that field consider much longer voltage-hold times [e.g., *J. Power Sources* 523, 231021 (2022)]. Are 50 hours just a starting point for coating studies?

(I think the authors are actually better off using the word "self-discharge" (the usage in silicon anode field) rather than "corrosion.")

My other outstanding objection is point (1) in my report, which the authors barely addressed. I will only support the publication of this work if the authors discuss the relation (similarity/difference) between typically (structural metal, say steel) corrosion and lithium corrosion. They should cite at least one paper on structural material corrosion, even if it means they have to drop a battery citation due to journal limits.

The rationale is the following. *Nature Communication* is a general audience journal. There are many readers interested in corrosion (which is a vibrant and important electrochemical topic). The authors use the word "corrosion" close to 80 times, but seem uninterested in its meaning, classification, etc., outside the battery community. If that's the case, if they are not interested in reaping multidisciplinary value and interest out of this, they may as well publish in more specialized battery journals like *Journal of Power Sources*.

I recommend they acquire more of a basic understanding of what corrosion means in other corrosion fields, what techniques are used in studying structural corrosion, what the main conceptual issues are. One example: many structural metals corrode via pitting only at overpotentials, because their native oxides are stable in electrolytes (i.e., their "SEI" do not readily "dissolve"). The oxides are largely impervious to cation diffusion. None of that describes lithium at room temperature, and indeed there is hardly any overpotential measured during lithium "corrosion" [*J. Phys. Chem. C* 126, 8565 (2022)], leading to the question whether true "galvanic corrosion" occurs there. I think adding that kind of cross-disciplinary insight, instead of using "corrosion" as a battery insider's buzzword that confuses the general, non-battery reader, is especially welcome for *Nature Communications*.

Reviewer #3 (Remarks to the Author):

The authors have sufficiently satisfied most of my comments.

The incorporation of an indirect test of SEI solubility improved the manuscript.

While other reviewers raise concerns about the use of the word corrosion to describe anode dissolution and SEI growth, there should be no such concern. The use of this term in the context of Li metal, SEI growth, and reaction of Li with liquid electrolyte can be traced at least as far back as the seminal 1979 paper from Peled. The SEI is a passivation film that kinetically limits further corrosion of the Li, which is caused by parasitic reduction of the electrolyte.

Point-by-point response to the referee questions

Reviewer #1 (Remarks to the Author):

Authors significantly improved quality of the revised manuscript addressing comments mentioned in the 1st review. Therefore, I would like to recommend the manuscript for publishing in Nature Communications after addressing all points described below:

Response: We highly appreciate the reviewer for the professional and valuable comments, which greatly help us improve the work to fit the high scope of *Nature Communications*. We have carefully checked your new comments, and tried our best to addressing your remaining concerns in the revised manuscript and response letter point-by-point. Thank you again for your precious time and approval of our work.

Comment 1: All text including title: Despite demonstration of certain SEI dissolution, I would not attribute the observed improvement only by this factor since literature indicates many other causes. I think the interface protection of Li anode should be mentioned in more broad discussion because, as it was proven before, irreversible Li loss have several causes. Moreover, conceptually speaking, Li metal will always have SEI layer in corresponding electrolytes. IMO, this simple argument clearly shows limitation of 100% focus of authors on the corrosion.

Response: Thank the reviewer for raising such a detailed and suggestive comment. We agree that many causes together are responsible for the irreversible Li loss, and Li metal always has a SEI layer. Referring to your recommendation, we have described the different causes for Li loss, and supplement more discussions on the multiple functions of the shield, which help improve the quality of this work.

Typically, Li loss can be triggered by: 1) Incomplete Li stripping and volumetric changes; 2) Chemical/electrochemical corrosion of Li; 3) Formation and dynamic evolutions of SEI. Notably, **SEI and its nature highly relate to the irreversible Li loss**. On one hand, the formation and dynamic chemical/structural evolution of SEI during battery operation or storage induce direct Li loss. On the other hand, instable

SEI can trigger Li corrosion and formation of dead Li, leading to the increase of irreversible Li loss. In this work, we have revealed the relation among Li loss, SEI dissolution and Li corrosion mainly by electrochemical protocols and cryo-TEM. Li corrosion and SEI dissolution have been validated as the important contributors to the irreversible Li loss.

Fig. R1 | Capacity loss caused by SEI formation, incomplete stripping and Li corrosion.

With the purpose of better understanding Li corrosion and interfacial protection, the contribution of different processes to the irreversible Li loss has been primarily quantified in Fig. R1. The capacity loss without resting treatment indicates the Li loss coupled with SEI formation and incomplete stripping process (~69 µAh), which matches well with the previous report in the same cell configuration (55 µAh)¹. Notably, Li loss from SEI formation (Li⁺) was decided to be ~39 µAh, while Li loss from incomplete Li stripping (Li⁰) was ~16 µAh. As can be seen in Fig. R1, Li loss from Li corrosion is comparable to that from the incomplete Li stripping and SEI formation. With the increase of resting time, Li corrosion plays a dominant role in Li loss (~75 µAh for 300 hours). Additionally, SEI dissolution also directly contributes to a certain amount of Li loss (~17 µAh for 50 hours, Supplementary Fig. S3c), whose major role

is to trigger the exposure of fresh Li and the subsequent Li corrosion. Such results validate the significance of resolving the issues of SEI dissolution and Li corrosion, especially for the prolonged operation of batteries. In our work, Li corrosion was successfully suppressed with the adoption of a prototypical shielding strategy (reduction by 74%). Notably, the shield exhibits multiple functions, which act together to reduce the irreversible Li loss: 1) The shield can prevent the SEI dissolution and Li corrosion by isolating electrolyte penetration; 2) The in-situ formed Li-Mg alloy and LiF at the anode surface deliver superior corrosion resistance to inhibit corrosion-induced Li loss; 3) The designed shield can also promote the uniform ion transportation and Li deposition, suppressing the formation of Li dendrite and dead Li, subsequently reducing the irreversible Li loss.

In the revised manuscript, we have provided more broad discussion on the interface protection. The multiple functions of the shielding layer have been discussed to avoid the 100% focus on the corrosion. The changes of the above aspects have been supplemented in the revised manuscript highlighted with yellow, which can also be found below.

“an anti-corrosion and interface-stabilizing shield comprising of low-solubility polymer and metal fluoride is designed is designed”. (Page 1, Line 20 and 21)

“However, stability and lifespan of Li metal battery still remain formidable challenges, which highly relate to the irreversible Li loss associated with incomplete Li stripping, interface evolutions, and corrosion behaviors.” (Page 2, Line 10–13)

“The Li corrosion and SEI dissolution emerge as additional failure mechanisms of reactive Li metal anodes”. (Page 3, Line 4 and 5)

“Notably, cells with a resting time within 50 hours delivered rapidly enlarged capacity loss of Li deposits and deteriorated CE (125 μ Ah, and 89% for 50 hours). With the increasing of resting time, Li corrosion continues to induce the increase of capacity loss and deterioration of CE (170 μ Ah, and 85% for 500 hours), while the changes become much slower”. (Page 4, Line 15–20)

“The amorphous polymeric outer layer containing PVDF inhibited Li corrosion, while

the inner inorganic layer comprised of Mg- and Li-containing species (e.g., Mg, LiMg, Li₂O, LiOH) promoted the rapid ion transportation at the interface". (Page 7, Line 6–9)

"In terms of the mixed ionic and electronic conducting layer of Mg- and Li-containing species, it promotes the fast ion transportation and uniform Li deposition". (Page 7, Line 16 and 17)

"Such improvement of CE potentially relates to the uniform Li deposition and reduced corrosion guided by Li-Mg alloy and LiF at the interface⁴⁶⁻⁴⁹". (Page 7, Line 29 and 30)

"These passive layers are validated to effectively regulate the nucleation and the subsequent Li plating/stripping processes. The Li | Cu and Li | Li cells were enabled to deliver stable cyclability, large CE, and low voltage hysteresis, indicating a stable Li metal–liquid electrolyte interface, suppressed Li corrosion, and fast ion transportation". (Page 8, Line 13–17)

"The PVDF matrix enables the isolation of Li metal and native SEI from electrolyte, while the fluorides react with Li to form Li alloy and LiF to further suppress the Li corrosion. Additionally, the shield successfully facilitates the ion transportation and deposition, preventing the dendrite growth and dead Li accumulation". (Page 9, Line 27–31)

Comment 2: Line 203. As it was mentioned by one of the reviewers, please do not exaggerate some points. For example, "corrosion-free" which is very brave and unsupported claim.

Response: Thank the reviewer for this detailed suggestion. We agree that the corrosion cannot be totally eliminated, even with the adoption of shields. The term "corrosion-free" has been replaced by "reduced corrosion". We have carefully checked the whole text to avoid and correct such inappropriate descriptions in the revised manuscript. The corresponding changes can also be observed below.

"enabling the durable operation of Li metal batteries by reducing the Li loss". (Page 1, Line 25)

“Such a dilemma highly relates to the limited recognitions on the Li corrosion science, and Li loss still undergoes³³⁻³⁵, appealing for more expeditions to uncover the correlation between dynamic Li corrosion and SEI progression (Supplementary Table S2). The corrosion science in battery systems needs to be understood and established (Supplementary Notes 1-4 and 1-5)”. (Page 3, Line 11–15)

“promising the reduced corrosion of Li deposits”. (Page 7, Line 15)

“the Li deposits can maintain the integral morphology after being soaked in the electrolyte”. (Page 7, Line 33; Page 8, Line 1)

Comment 3: Line 267. “full cell” appears again. Please check.

Response: We highly appreciate the reviewer for the reminding. We have carefully checked the whole text to correct such descriptions. The changes have been highlighted with yellow in the revised manuscript, which can also be found below.

“Prolonged operations of Li symmetric cells and LiFePO₄ | Li cells”. (Page 1, Line 21 and 22)

“Electrochemical evaluations of cells comprising of Li and commercial cathodes”. (Page 8, Line 20)

“the cell configuration paired with commercial LiFePO₄ cathode were assembled and evaluated”. (Page 8, Line 21 and 22)

“The evaluations of Li | LiFePO₄ and Li | NCM523 cells”. (Page 9, Line 12 and 13)

“The Li | LiFePO₄ cells were composed of LiFePO₄ cathodes”. (Page 12, Line 7)

“Fig. 5 | Cell evaluations”. (Page 22, Line 1)

Comment 4: Conclusions: I think this part should be improved via providing more concrete results (for example, the best chemistry for protection layer).

Response: Thank the reviewer for raising such a professional and significant question, which really helps us improve the quality and value of this work. In the conclusion section, we have supplemented more concrete results and prospects on the protective layer in the revised manuscript highlighted with yellow, which has also been presented

below.

“A prototypical strategy is to introduce a protective layer to inhibit the SEI dissolution and Li corrosion by electrolyte. The matrix of such a shield should be stable against electrolyte, Li metal, and interfacial stress (e.g., low solubility, reductive stability, flexibility, mechanical stability), while the filler inside the shield can promise the uniform ion transportation and Li deposition (e.g., ionic conductivity, lithiophilicity). Notably, the composite layer comprised of polymer and fluorides (e.g., MgF₂@PVDF) can serve as a typically ideal passive shield to largely reduce the corrosion of reactive anodes like Li metal by 74%. The PVDF matrix enables the isolation of Li metal and native SEI from electrolyte, while the fluorides react with Li to form corrosion-resistant Li alloy and LiF to further suppress the Li corrosion. Additionally, the shield successfully facilitates the ion transportation and deposition, preventing the dendrite growth and dead Li accumulation”. (Page 9, Line 19–31)

“The findings and strategy can further be extended to other battery systems using reactive metallic anodes (e.g., Na, K, Zn), and promote the development of corrosion science in battery systems”. (Page 10, Line 1–3)

Comment 5: Figure 5f: I think this drawing should be removed.

Response: Thank the author for this suggestion, and we have deleted Figure 5f in the revised manuscript. The revised figure can also be found below (**Fig. R2**).

Fig. R2 | Cell evaluations in pair with LiFePO₄ cathode. **a** Rate capability of Li | LiFePO₄ cells with and without a MgF₂@PVDF shield at 0.1, 0.2, 0.5, 1, 2, 5, 10, and 1C. **b, c** Voltage profiles of Li | LiFePO₄ cells without and with a MgF₂@PVDF shield, respectively. **d, e** Prolonged cycling performance of Li | LiFePO₄ cells with and without a MgF₂@PVDF shield at 1C, and the corresponding CE. **f** Digital photo of a MgF₂@PVDF-Li | LiFePO₄ pouch cell. **g** Cycling performances of Li | LiFePO₄ pouch cells with and without a MgF₂@PVDF shield.

Reviewer #2 (Remarks to the Author):

The authors have extensively revised their manuscript and added evidence of SEI dissolution. They have also changed their abstract so it no longer claims to show proof of SEI dissolution causing Li "corrosion" (it can easily be the other way round) -- that is much appreciated. While Referee 1 helpfully pointed out several review articles documenting PVDF/MgF₂ as a coating agent, it is true, as the authors point out, that most of the work reviewed there is not related to Li "corrosion."

Overall I now recommend publishing this work provided the two points below are addressed (not peripherally, but in some detail).

Response: We are grateful for the kind supports and suggestions from the reviewer, which really help us improve the quality and significance of this work to the broad readers of Nature Communications. We have carefully checked your two valuable questions. Necessary experimental results and detailed discussion on corrosion are provided in the revised manuscript and supplementary information, involving the typical corrosion research and corrosion in battery systems.

(1) In terms of comment 1, we have extended the time scale of resting to 300 and 500 hours referring to the paper on silicon anodes. The result indicates the continuous Li corrosion, which induces rapid increase of capacity loss of Li deposits within the initial 50 hours, and gradual increase of Li loss subsequently (from 50 to 500 hours).

(2) To response to comment 2, we have provided discussion on relation between structural metal corrosion and Li corrosion: 1) The concept and classification of corrosion; 2) Characterization techniques in corrosion science; 3) Issues and topics in corrosion science; 4) The similarities and differences between typical corrosion and Li corrosion; 5) The significance of corrosion science in battery systems.

Point-by-point response can be found below, and we hope our responses can well resolve your concerns and questions. Thank you again for your professional comments and precious time.

Comment 1-1: One issue the authors have not raised is the self-discharge time scale

involved. The authors go to ~50 hours; is this sufficient? One of the reason silicon anodes continue to struggle to be commercialized is their limited calendar life, which is related to self-discharge rate. Researchers in that field consider much longer voltage-hold times [e.g., *J. Power Sources* 523, 231021 (2022)]. Are 50 hours just a starting point for coating studies?

Response: We sincerely thank the reviewer for such a valuable issue. We have carefully checked the related papers, and extended the time scale of resting to 300 and 500 hours (Fig. R3a). As can be seen below, Coulombic efficiency decays sharply during the initial 50 hours, and the irreversible capacity loss relating to Li corrosion rapidly accumulates (Fig. R3b, c). Note that both the decrease of efficiency and the increase of capacity loss become slower with the increase of resting time due to the growth or stacking of SEI. **The supplemented results indicate the continuous Li corrosion during a prolonged time scale, which should be carefully taken into account for the battery research.** Besides, such findings are similar with the reported result by He *et al.*, claiming the major corrosion behaviors of Li undergoes within the initial 100 hours². We have made changes to the corresponding figures (Figs. 1f, 3a, and Supplementary Fig. S3a) and descriptions in the revised manuscript and supplementary information, which can also be found below (Fig. R3).

Fig. R3 | Corrosion test. **a** Voltage profiles of pristine Li | Cu cells with different resting time for detecting Li corrosion. **b, c** Corrosion related capacity loss of Li and the corresponding CE as a function of rest time in Li | Cu cells without and with a shield. The cell was cycled at 1.0 mA cm^{-2} and room temperature with a Li cycling capacity of 1.0 mAh cm^{-2} .

“while after a duration of rest (0, 5, 15, 50, 300, and 500 hours) Li was totally stripped (Supplementary Fig. S3a)”. (Page 4, Line 10 and 11)

“Notably, cells with a resting time within 50 hours delivered rapidly enlarged capacity loss of Li deposits and deteriorated CE (125 μ Ah, and 89% for 50 hours). With the increasing of resting time, Li corrosion continues to induce the increase of capacity loss and deterioration of CE (170 μ Ah, and 85% for 500 hours), while the changes become much slower”. (Page 4, Line 15–20)

“With the increase of resting time, the capacity loss of Li deposits shows gradual increase (84 μ Ah for 500 hours)”. (Page 6, Line 16 and 17)

“The findings and strategy can further be extended to other battery systems using reactive metallic anodes (e.g., Na, K, Zn), and promote the development of corrosion science in battery research”. (Page 10, Line 1–3)

Comment 1-2: I think the authors are actually better off using the word "self-discharge" (the usage in silicon anode field) rather than "corrosion."

Response: We thank the author for this kind suggestion. All the authors have carefully thought about this great question. After that, we tried to provide our reasons and concerns for such a prior selection of “corrosion” in this work, hoping to get your consensus and approval. In fact, the terms “self-discharge” and “corrosion” are both used in battery community and closely related, while corrosion distinguishes from self-discharge in several aspects. The former is mainly applied to describe the battery-level energy loss, while the latter focuses on the material-level interfacial changes of anode. In this work, we are devoted to validating Li corrosion and SEI dissolution at the anode side, and consequently the term “corrosion” is selected and used. The detailed reasons why the term “corrosion” instead of “self-discharge” is adopted in our work have been concluded as follows:

1) **Self-discharge, a battery-level phenomenon, is often used to describe the natural loss of charge capacity of batteries at state of charge (SOC)**, which can be accelerated in a higher SOC³. For instance, the loss of Li inventory from lithiated

electrodes (*e.g.*, lithiated Si)⁴ lead to the self-discharge, which challenges the calendar life of batteries. In contrast, **Li corrosion is material-level direct conversion of Li⁰ to Li⁺ at the anode side, which undergoes during battery charging/discharging and storage without considering the SOC.** Besides, there are irreversible and reversible (*e.g.*, soft short circuits, transition metal dissolution) forms of self-discharge⁵, while corrosion conversely is highly irreversible. In this work, we systematically studied the Li corrosion at the anode side, which provides extra explanation to irreversible Li loss and battery failure. Notably, the relation among Li corrosion and SEI dissolution has been revealed.

2) **Many factors contribute to the self-discharge of batteries,** including the internal/external electron leakage, electrode/electrolyte reactions, partial dissolution of active material, and electrode mechanical disintegration⁶. Li corrosion caused by electrode/electrolyte parasitic reactions is one such typical factor relating to the self-discharge of batteries. Consequently, using “self-discharge” (the result) to replace “corrosion” (the cause) may be inaccurate. In this work, with the assistance of cryo-TEM and electrochemical protocols, the SEI dissolution and Li corrosion were visualized and quantitatively confirmed by monitoring the corroded Li and dissolved SEI.

3) **Li corrosion is a combination of corrosion science and battery chemistry,** which shows similarities with corrosion of typical structural metals. For instance, the structure, chemistry, and function of metals change and deteriorate due to the corrosion reactions between the metals and reactive environments. Additionally, a passive layer of corrosion products forms on both kinds of metals. Moreover, corrosion of Li metal and structural metals both proceeds due to the breakage/failure of passive layers. Consequently, it's appropriate and significant to use “corrosion” in this work, which to some extent contributes to the development of corrosion science in battery systems.

4) Corrosion in battery systems is an important part of both battery and corrosion research. In battery systems, various patterns of corrosion can be found in different components like current collector (*e.g.*, pitting of Al), electrode (*e.g.*, intergranular corrosion), and solid-state electrolytes (*e.g.*, stressing corrosion crack)^{7,8}. As

Reviewer#3 mentioned, **the term of corrosion has been adopted in battery community since 1979**. Peled *et al.* described the SEI as the corrosion products from parasitic reaction between Li and electrolyte, which pioneeringly introduced the corrosion science into Li-based battery systems^{9,10}. Herein, corrosion of Li metal anode is studied to reveal its influence on battery performances, and relevant strategy for solving such issues are proposed. Consequently, corrosion is preferentially used in this work.

On the basis of the above concerns, “corrosion” is selected and used, which helps indicate the significance of developing corrosion science in battery systems.

Comment 2-1: My other outstanding objection is point (1) in my report, which the authors barely addressed. I will only support the publication of this work if the authors discuss the relation (similarity/difference) between typical (structural metal, say steel) corrosion and lithium corrosion. They should cite at least one paper on structural material corrosion, even if it means they have to drop a battery citation due to journal limits.

Response: Thank the reviewer for this valuable comment. We have carefully read the papers on corrosion of typical structural metals, and concluded the relation (similarity/difference) between typical corrosion and Li corrosion. In addition, several papers on structural metal corrosion have been included in the revised manuscript. Such changes are highlighted with yellow in the revised manuscript and supplementary information to increase the multidisciplinary value of this work, which can also be observed below.

“The similarities and differences between typical corrosion and Li corrosion

Similarity: 1) Li like typical structural metals can undergo various kinds of corrosion due to the interaction between metals and reactive environments, including pitting, stress corrosion, intergranular corrosion, and galvanic corrosion in contact with a noble metal^{18,19}; 2) Corrosion of Li and structural metals will trigger the changes/deterioration of their structures, chemistries, and functions; 3) Corrosion

leads to the formation of passive layers on both metal surfaces, which is comprised of corrosion products; 4) The corrosion of Li and structural metals both relates to the breakage/failure of passive layers. For example, stainless steels are susceptible to pitting corrosion due to the localized dissolution of an oxide-covered metal in aggressive environments²⁰. In terms of the passive layer of Li (i.e., SEI), it suffers mechanical and chemical instability. Due to the cracks and dissolution of SEI, Li exposed to the electrolyte will be continuously corroded; 5) The corrosion prevention strategies in typical corrosion science can be successfully translated into battery systems, including the coating, doping, and regulation of the alloy compositions; 6) The characterization techniques of corrosion research can also be adopted in battery systems, such as the electron backscatter diffraction.

Difference: 1) Overpotentials are less likely to occur during Li corrosion due to the extremely low redox potential of Li and sufficiently fast Li⁺ diffusion through the SEI²¹. Conversely, corrosion of structural metals usually involves overpotentials against metal plating, spatial segregation of cathodic and anodic processes, and kinetics considerations²²; 2) The compositions and functions of passive layers on Li and structural metals are different. The passive layer of structural metals formed in atmospheric environments or aqueous solutions mainly consists of inorganic oxide or hydroxide, some of which can suppress the proceeding of corrosion (e.g., Al₂O₃ on Al), while others cannot inhibit the corrosion (e.g., rust on steel). In contrast, the passive layer on Li (i.e., SEI) usually comprises of organic and inorganic Li salts, which is supposed to inhibit the continuous corrosion reactions, meanwhile enabling the reversible operation of batteries (i.e., charging/discharging, or plating/stripping); 3) Li with high redox power, serious volumetric effect, and instable SEI will be repeatedly exposed to the electrolyte, making the prevention of Li corrosion much more challenging than that of structural metal; 4) Since Li is sensitive to heat, electron, and air, expeditions on Li corrosion are faced with greater difficulty than that of the structural metal corrosion. The investigation of Li corrosion depends more tightly on advanced characterization techniques.” (Supplementary Note 1-3, Page S4 and S5)

“Corrosion is an irreversible system phenomenon of materials (e.g., structural metal, ceramic, and polymer)¹³⁻¹⁵, involving such diverse factors as material, coating, environment, microbiology, stresses and/or electromagnetism (Supplementary Notes 1-1, 1-2, and Supplementary Table S1). Typically, the corrosion of a metal refers to the oxidation of the metal to its ionic species and release of electrons (e.g., rusting of steel), which challenges the longevity, safety and function of products (Supplementary Note 1-3)¹⁶. Li as a reactive metal is highly susceptible to corrosion, which typically is independent of extra external current or potential polarization, and contributes to the self-discharging of batteries^{17,18}. The terminology of corrosion in battery research can date back to 1979, when Peled et al. described the solid-electrolyte-interphase (SEI, i.e., a layer of corrosion product) at the Li metal–liquid electrolyte interface¹⁹”. (Page 2, Line 14–25)

“Similar with the corrosion of structural metals^{20,21}.” (Page 2, Line 25 and 26)

“Such a dilemma highly relates to the limited recognitions on the Li corrosion science, and Li loss still undergoes³³⁻³⁵, appealing for more expeditions to uncover the correlation between dynamic Li corrosion and SEI progression (Supplementary Table S2). The corrosion science in battery systems needs to be understood and established (Supplementary Notes 4 and 5)”. (Page 3, Line 11–15)

“13Eswarappa Prameela, S. et al. Materials for extreme environments. *Nat. Rev. Mater.* 8, 81–88 (2022).

14 Ryan, M. P., Williams, D. E., Chater, R. J., Hutton, B. M. & McPhail, D. S. Why stainless steel corrodes. *Nature* 415, 770–774 (2002).

15 Fujii, T., Suzuki, M. & Shimamura, Y. Susceptibility to intergranular corrosion in sensitized austenitic stainless steel characterized via crystallographic characteristics of grain boundaries. *Corros. Sci.* 195, 109946 (2022).

17 Leung, K., Merrill, L. C. & Harrison, K. L. Galvanic corrosion and electric field in lithium anode passivation films: Effects on self-discharge. *J. Phys. Chem. C* 126, 8565–8580 (2022).

18 Malkowski, T. F. et al. Evaluating the roles of electrolyte components on the

passivation of silicon anodes. J. Power Sources 523, 231021 (2022).

20 Yan, C. et al. *Evading strength-corrosion tradeoff in Mg alloys via dense ultrafine twins. Nat. Commun. 12, 4616 (2021).*

21 Zhu, Q. et al. *Towards development of a high-strength stainless Mg alloy with Al-assisted growth of passive film. Nat. Commun. 13, 5838 (2022)*” (Page 14, Line 8–27)

Comment 2-2: The rationale is the following. Nature Communication is a general audience journal. There are many readers interested in corrosion (which is a vibrant and important electrochemical topic). The authors use the word "corrosion" close to 80 times, but seem uninterested in its meaning, classification, etc., outside the battery community. If that's the case, if they are not interested in reaping multidisciplinary value and interest out of this, they may as well publish in more specialized battery journals like Journal of Power Sources.

Response: We're grateful to the reviewer for the valuable and professional suggestions, which really help us increase the quality and significance of this work to the general audience of Nature Communications. As the reviewer mentioned, corrosion is an important topic in various fields (*e.g.*, industry, energy, transport, infrastructure, cultural heritage). We have tried to understand “corrosion” from different aspects to improve the multidisciplinary value of our work by supplementing discussions on the meaning, classifications, and prevention strategies of corrosion.

(1) **The concept of corrosion.** Corrosion is an irreversible system phenomenon of materials (*e.g.*, metal, ceramic, and polymer), where such diverse factors as material, coating, environment, microbiology, stresses and/or electromagnetism act together¹¹. The corrosion of a metal refers to the oxidation of the metal to its ionic species and release of electrons, which highly relates to the physicochemical interaction between the metal and environment. The corrosion of metals limits the longevity, safety and function of products, and even results in abrupt and catastrophic failure. One such typical case is the rusting of steel, where a red-brownish hydrated iron oxide layer forms.

(2) **The classification of Corrosion.** 17 representative types of corrosion have

been summarized in Table R1 (*e.g.*, uniform corrosion, pitting corrosion, galvanic corrosion, crevice corrosion, intergranular corrosion, biological corrosion.), where the features and prevention of each corrosion have also been provided. In this work, Li corrosion via pitting and galvanic corrosion has been studied and validated, which relates to the SEI dissolution, and contributes to the irreversible Li loss.

(3) **Corrosion prevention.** Various strategies have been proposed to inhibit the corrosion in typical corrosion researches, mainly including protective layers (*i.e.*, coating), structural/composition design (*e.g.*, addition of anti-corrosion element), environment regulation, cathodic protection, and anodic protection (Table R1). Such strategies can also be translated into the prevention of Li corrosion. Due to the complexity of material systems and operational environments (*e.g.*, microstructure, temperature, pressure), more and more corrosion types and mechanisms emerge, appealing to continuous focus and investigation. One such a scenario is corrosion in battery systems, where the corrosion mechanism and scientific prevention strategies remain to be further explored.

Table R1. Features and prevention of different types of corrosion

Types		Features	Preventions	Refs.
Uniform corrosion		The most common type of corrosion caused by chemical or electrochemical attacks over the entire surface area of the metal	Corrosion allowance, alloy, and coatings	12
Localized corrosion	Pitting corrosion	Localized dissolution of an oxide-covered metal in specific aggressive environments, resulting in small holes/pits	Cathodic protection, anodic protection, alloy, and selection of high-corrosion-resistance materials	13
	Crevice corrosion	Occurring in or directly adjacent to gaps or crevices on the surface of a metal	Crevices avoidance, cathodic protection, inhibitors, and coatings	14
	Filiform corrosion	Occurring underneath a coating and manifesting itself as threadlike fibers	Reducing the humidity, multiple coats, and proper alloy	15
	Deposition corrosion	A more noble metal depositing on the less noble metal to trigger pitting	Maintaining cleanliness, proper use of protective coatings/sealants, and addition of corrosion-resistant elements	16
Galvanic corrosion/Bimetallic corrosion		Occurring between two dissimilar metals that are in electrical contact and exposed to electrolyte	Sacrificial material, selective coating, and insulating the galvanic couple	17

Environmental cracking	Stress corrosion	Caused by a corrosive environment with residual mechanical stress on the material surfaces (e.g. , welding, heat treatments, and cold deformations)	Stress relief (e.g. , annealing), surface treatments (e.g. , shot peening), and avoiding design of sharp corners, holes, etc. in materials	18
	Corrosion fatigue	Caused by a combination of cycling stress and corrosive environments	Alloys, coatings, inhibitors, reducing stress raisers, modifying the corrosive environment, and reducing residual stresses	19
	Hydrogen-induced cracking	Occurring in carbon or low-alloy steels when hydrogen atom diffuses into the materials and forms hydrogen molecules, internally pressurizing the material and initiating cracking	Reducing regions of high micromechanical contrast, designing dense oxide surface layers, and trapping hydrogen at proper internal interfaces and defects	20
Flow-assisted corrosion	Erosion corrosion	Triggered by the combined mechanical stress action from the fluid motion, and the corrosive chemical actions in the fluids	Using coatings and other surface treatment, and cathodic protection	21
	Cavitation corrosion	Caused by the formation and collapse of gas bubbles on the surface of the material, inducing initial cavitation	Ceramic/epoxy/polyurethane coatings, weld overlays, and plasma/thermal spray treatments	22

	Impingement corrosion	Localized erosion-corrosion caused by turbulence or impinging flow, damaging the protective oxide film	Corrosion-resistant coatings	23
Intergranular corrosion		A chemical or electrochemical attack on or adjacent to the grain boundaries of a metal	Conducting proper annealing and quenching treatments, and regulating the alloy compositions	24
Selective leaching corrosion/De-alloying		Preferential removal of one constituent of an alloy to leave behind an altered residual structure (e.g. , de-zincification of brass)	Using cladding process, adding proper element, and cathodic protection	25
Fretting corrosion		Caused by small cyclic movements between two materials associated with corrosive attack from the environment	Reducing the relative motion, increasing the interfacial stability, and using anti-fretting coatings, shot peening, and lubrication	26
High-temperature corrosion		A chemical attack from gases, solid or molten salts, or molten metals at high temperatures	Overlay and thermal barrier coatings	27
Biological corrosion		Caused by biological organisms or microorganisms	Coatings, cathodic protection, biocides, and cleaning procedures	28

The supplementary contents on the conception and classification of corrosion have been added in the revised supplementary information highlighted with yellow, which have also been presented below.

“I-1. Features and prevention of different types of corrosion

Uniform corrosion. The most common type of corrosion caused by chemical or electrochemical attacks over the entire surface area of the metal¹.

Solutions: Corrosion allowance, alloy, and coatings.

Localized corrosion

Pitting corrosion. Localized dissolution of an oxide-covered metal in specific aggressive environments, resulting in small holes/pits².

Solutions: Cathodic protection, anodic protection, alloy, selection of high-corrosion-resistance materials.

Crevice corrosion. Occurring in or directly adjacent to gaps or crevices on the surface of a metal³.

Solutions: Crevices avoidance, cathodic protection, inhibitors and coatings.

Filiform corrosion. Occurring underneath a coating and manifesting itself as threadlike fibers⁴. Solutions: Reducing the humidity, and multiple coats, proper alloy.

Deposition corrosion. A more noble metal depositing on the less noble metal to trigger pitting⁵.

Solutions: Maintaining cleanliness, and proper use of protective coatings/sealants, and addition of corrosion-resistant elements.

Galvanic corrosion/Bimetallic corrosion. Occurring between two dissimilar metals that are in electrical contact and exposed to electrolyte⁶.

Solutions: Sacrificial material, selective coating, insulating the galvanic couple.

Environmental cracking

Stress corrosion. Caused by a corrosive environment with residual mechanical stress on the material surfaces (e.g., welding, heat treatments and cold deformations)⁷.

Solutions: Stress relief (e.g., annealing), surface treatments (e.g., shot peening), and avoiding design of sharp corners, holes, etc. in materials.

Corrosion fatigue. Caused by a combination of cycling stress and corrosive environments⁸.

Solutions: Alloys, coatings, inhibitors, reducing stress raisers, modifying the corrosive environment, and reducing residual stresses.

Hydrogen-induced cracking. Occurring in carbon or low-alloy steels when hydrogen atom diffuses into the materials and forms hydrogen molecules, internally pressurizing the material and initiating cracking⁹.

Solutions: Reducing regions of high micromechanical contrast, designing dense oxide surface layers, and trapping hydrogen at proper internal interfaces and defects

Flow-assisted corrosion

Erosion corrosion. Triggered by the combined mechanical stress action from the fluid motion, and the corrosive chemical actions in the fluids¹⁰.

Solutions: Using coatings and other surface treatment, and cathodic protection.

Cavitation corrosion. Caused by the formation and collapse of gas bubbles on the surface of the material, inducing initial cavitation¹¹.

Solutions: Ceramic/epoxy/polyurethane coatings, weld overlays, and plasma/thermal spray treatments.

Impingement corrosion. Localized erosion-corrosion caused by turbulence or impinging flow, damaging the protective oxide film¹².

Solutions: Corrosion-resistant coatings.

Intergranular corrosion. A chemical or electrochemical attack on or adjacent to the grain boundaries of a metal¹³.

Solutions: Conducting proper annealing and quenching treatments, and regulating the alloy compositions.

Selective leaching corrosion/De-alloying. Preferential removal of one constituent of an alloy to leave behind an altered residual structure (e.g., de-zincification of brass)¹⁴.

Solutions: Using cladding process, adding proper element, and cathodic protection.

Fretting corrosion. Caused by small cyclic movements between two materials associated with corrosive attack from the environment¹⁵.

Solutions: Reducing the relative motion, increasing the interfacial stability, and using anti-fretting coatings, shot peening, and lubrication.

High-temperature corrosion. A chemical attack from gases, solid or molten salts, or molten metals at high temperatures¹⁶.

Solutions: Overlay and thermal barrier coatings.

Biological corrosion. Caused by biological organisms or microorganisms¹⁷.

Solutions: Coatings, cathodic protection, biocides, cleaning procedures¹⁸.

(Supplementary Note 1-1, Page S2 and S3)

Comment 2-3: I recommend they acquire more of a basic understanding of what corrosion means in other corrosion fields, what techniques are used in studying structural corrosion, what the main conceptual issues are.

Response: Thank you for such professional and constructive comments. Referring to your suggestion, we have carefully checked the literature on corrosion, and supplemented more understanding on corrosion in different fields to enhance the significance of this work. The characterization techniques for uncovering corrosion mechanisms and migration strategies have been summarized. In addition, emerging issues and topics in corrosion science have also been concluded. Such changes have been added in the revised supplementary information highlighted with yellow, which can also be found below (Table R2).

(1) **Characterization techniques in corrosion science.** Typical characterization techniques in materials science on microstructure and chemistry examination can be successfully translated into the study of corrosion science. For instance, focused ion beam-scanning electron microscopy (FIB-SEM) accompanied by electron backscattered diffraction (EBSD) has been frequently used to visualize the corrosion morphology and crystal orientation of corroded metals. Yang *et al.* used FIB-SEM to create a 3D reconstruction of sample to prove the connectivity of the corroded voids along the grain boundaries²⁹. Zhao *et al.* by atom probe tomography (APT) performed near-atomic-scale analysis of H trapped in second-phase particles and at grain boundaries in a high-strength Al alloy, revealing the prevention of H embrittlement³⁰. Moore *et al.* used a high-speed atomic force microscopy (HS-AFM) to detect the in-

situ stress corrosion cracking on a stainless steel in an aggressive salt solution³¹. Cryo-transmission electron microscopy (cryo-TEM) is an emerging technique in material science, which have been used in both corrosion science and battery researches. Such a technique can atomically visualize the structural and chemical information of passive layers and corroded metals^{32,33}. For instance, Lin *et al.* investigated a corroded NiCrMo alloy in a frozen-hydrated state during aqueous corrosion with the assistant of cryo-TEM³².

Additionally, **electrochemical measurements are also important techniques for quantitative corrosion researches**. For example, Mouanga *et al.* employed the localized electrochemical impedance spectroscopy (LEIS) to study the galvanic corrosion between zinc and carbon steel when exposed to a dilute NaCl solution³⁴. Manhabosco *et al.* used scanning vibrating electrode technique (SVET) to verify the increased active dissolution (*i.e.*, corrosion rate) of laser marking on M340 martensitic stainless steel due to Cr-depleted regions³⁵. Zero resistance ammetry (ZRA) technique is an efficient tool for quantifying the corrosion current/rate. For instance, Miller *et al.* employed ZRA to investigate the microbially influenced corrosion of carbon steel³⁶. Last but not the least, **computational and artificial intelligence are promising in the exploration and investigation of corrosion science**³⁷.

Table R2. Techniques for exploring corrosion science (Supplementary Table S1, Page S38 and S39)

Characterization techniques	Features	Refs.
Microstructure and chemistry examination		
Focused ion beam-scanning electron microscopy (FIB-SEM)	Visualizing the corrosion morphology on the surface and inside the materials	29
Cryo-electron transmission microscopy (cryo-TEM)	Atomically visualizing the nanostructure and chemistry of passive layer and corroded metals	32
High-angle annular dark-field scanning transmission electron microscopy (HAADF-STEM)	Detecting the atomic-resolution fine structures of corroded materials (e.g. , dislocations)	38
Electron backscatter diffraction (EBSD)	Measuring the grain orientation and size distribution of corroded materials	39

In-situ optical microscopy	Monitoring the corrosion behaviors and evolution process of metal surfaces	40
Atom probe tomography (APT)	Providing 3D compositional mapping of materials with sub-nanometer spatial resolution and no elemental mass limits	41
High-speed atomic force microscopy (HS-AFM)	Measuring stress corrosion cracking with high temporal and spatial resolution	31
Electrochemical measurement		
Scanning vibrating electrode technique (SVET)	Recording the local current density mappings and potentiodynamic gradients at metal/alloy surface	42
Kelvin probe force microscopy (KPFM)	Mapping the work function or surface potential of the sample with high spatial resolution	43
Localized electrochemical impedance spectroscopy (LEIS)	Measuring the local kinetics, capacitance, and resistance	44
Scanning electrochemical microscopy (SECM)	Monitoring the electron transfer with high spatial resolution, enabling the comprehension of corrosion mechanisms and mitigation strategies	45
Potentiodynamic polarization (PDP) method	Obtaining the corrosion potential to reveal the nobility and corrosion current of metals/alloys, indicating the corrosion rate	46
Zero resistance ammetry (ZRA) technique	Quantifying the galvanic coupling current between two dissimilar electrodes	47

“Typical characterization techniques in materials science on microstructure and chemistry examination can be successfully translated into the study of corrosion science. Additionally, electrochemical measurements are also important techniques for quantitative corrosion researches. The techniques for corrosion research have been summarized in Supplementary Table S1. Last but not the least, computational and artificial intelligence are promising in the exploration and investigation of corrosion

science". (Supplementary Note 1-2, Page S3)

(2) **Issues and topics in corrosion science.** "*The issues and topics in corrosion science presently include^{11,48}: 1) Enhancing the strength and corrosion resistance of a material concurrently. The high strength of materials mainly comes from complex microstructures, chemical compositions and internal stress fields, which make such materials susceptible to corrosion; 2) Increasing the sustainability/recyclability of alloys and maintaining the corrosion resistance; 3) Designing advanced corrosion inhibitors, coatings, and cathodic protection techniques; 4) Developing ab initio-based computational and artificial intelligence in corrosion science; 5) Designing analytical and in-operando methods for high-resolution and quantitative corrosion probing, especially for hydrogen induced corrosion; 6) Developing hydrogen-embrittlement-resistant materials for the use of hydrogen as an energy carrier (e.g., doped medium- and high-entropy alloy variants); 7) Corrosion in the oil and gas industry, nuclear facilities, and battery. Notably, the corrosion in battery systems is thought to be an important topic for both corrosion science and battery research*". (Supplementary Note 1-4, Page S5)

Comment 2-4: One example: many structural metals corrode via pitting only at overpotentials, because their native oxides are stable in electrolytes (i.e., their "SEI" do not readily "dissolve"). The oxides are largely impervious to cation diffusion. None of that describes lithium at room temperature, and indeed there is hardly any overpotential measured during lithium "corrosion" [J. Phys. Chem. C 126, 8565 (2022)], leading to the question whether true "galvanic corrosion" occurs there.

Response: We thank the reviewer for this professional question, which inspires us to supplement more discussion on corrosion from an interdisciplinary perspective. In terms of typical structural metals, the frequently observed oxide-based passive layer is relatively stable and ionically insulating. Cation diffusion through such oxide layers is sluggish, and becomes the corrosion-limiting step⁴⁹. Consequently, the proceeding of corrosions need to be driven by an external potential polarization to break down the passive layer or accelerate the ion diffusion. In sharp contrast, no extra external current or potential polarization is needed for galvanic corrosion of Li. In batteries using Cu as the current collector or anode, a galvanic cell forms, where active Li serves as the anode

and noble Cu as the cathode, triggering the namely bimetallic corrosion (*i.e.*, galvanic corrosion). Notably, the galvanic corrosion of Li refers to the oxidation of Li metal, and reduction of electrolyte on Cu surface. On one hand, SEI on Li metal suffers dissolution and breakage, and leads to the repeated exposure of Li to the electrolyte. Moreover, exposed Li metal with high redox power can intermediately react with electrolyte upon contact. On the other hand, the passive layer on Cu surface is reported to be porous, organic-rich, and ionically conductive, which promises the sufficiently fast Li⁺ diffusion, electron tunnelling, and charge transfer⁵⁰. Consequently, no extra overpotential is needed for Li corrosion. Additionally, galvanic corrosion of Li has been validated by several groups. For instance, Cui's group designed a galvanic cell comprised of Cu and Li that was externally connected by an ammeter, where the corrosion current ($\mu\text{A cm}^{-2}$ level) was recorded to indicate the corrosion rate⁵⁰. Ding *et al.* studied the dynamic galvanic corrosion in practical batteries during Li stripping⁵¹.

Comment 2-5: I think adding that kind of cross-disciplinary insight, instead of using "corrosion" as a battery insider's buzzword that confuses the general, non-battery reader, is especially welcome for Nature Communications.

Response: We sincerely thank the reviewer for the helpful suggestion. We have supplemented more discussion to reveal the relation between Li corrosion and typical corrosion of structural metals (responses to comments 2-1, 2-2, and 2-3), with the purpose of increasing cross-disciplinary insights for the broad readers in Nature Communications. Li corrosion is a specific kind of corrosion. Researches on such an interdisciplinary topic contribute to the development of both corrosion science and battery chemistry, the significance of which can be found below.

“The significance of corrosion science in battery systems. Battery materials including electrodes, current collector, and solid-state electrolytes are susceptible to corrosion, which largely deteriorate the battery performances and should be seriously considered. Notably, corrosion in batteries exhibits the typical features of corrosion, while it also distinguishes from the corrosion of structural metals, especially for Li metal anode of ultra-active nature and complex SEI. The synergistic investigations of corrosion science and corrosion prevention in Li metal batteries will yield interdisciplinary benefits to both fields: 1) New recognition on battery failure mechanism (hydrogen, stress, etc. related corrosion); 2) Guiding the design and strategies for batteries with longevity; 3)

Inspiring the corrosion prevention in other fields; 4) Promoting the development of advanced characterization techniques; 5) Extending and establishing corrosion science in battery systems". (Supplementary Note 1-5, Page S5 and S6)

With such supplemented contents and discussions, we believe this work can be improved to satisfy the broad interest of readers from different fields. And we hope our responses have addressed your concerns. Thank you again for your professional suggestions and continuous support.

Reviewer #3 (Remarks to the Author):

The authors have sufficiently satisfied most of my comments.

The incorporation of an indirect test of SEI solubility improved the manuscript.

While other reviewers raise concerns about the use of the word corrosion to describe anode dissolution and SEI growth, there should be no such concern. The use of this term in the context of Li metal, SEI growth, and reaction of Li with liquid electrolyte can be traced at least as far back as the seminal 1979 paper from Peled. The SEI is a passivation film that kinetically limits further corrosion of the Li, which is caused by parasitic reduction of the electrolyte.

Response: We thank the review very much for the approval of our revised work and professional suggestions, which greatly help us improve the quality of this work. And the valuable comment on the “corrosion” also inspires us to resolve the concerns and questions on this terminology.

As you mentioned, Peled in 1979 introduced the term “corrosion” in Li metal batteries, and SEI (*i.e.*, a passive layer) was described to be comprised of corrosion products¹⁰. The reasons why the term “corrosion” is used in our work can be concluded as: 1) Li corrosion is a specific kind of corrosion, which shows the typical features of corrosion; 2) Li corrosion similar with structural metal corrosion consists of the oxidation of metal, and reduction of electrolyte; 3) Corrosion will trigger the structural, chemical, and functional changes of Li metal and structural metals; 4) The proceeding of Li corrosion as well as the structural metal corrosion depends on the breakage/failure of passive layer; 5) Note that various corrosion forms can be found in a battery, such as the stress corrosion cracking of ceramic solid-state electrolyte and pitting of Al current collector^{52,53}. Developing corrosion science in battery systems is of certain significance, which can provide new insights on corrosion and battery researches.

More descriptions and discussions on Li corrosion and typical corrosion have been added in the revised manuscript and supplementary information highlighted with yellow, with the purpose of making this paper of higher multidisciplinary value. Such changes can also be found in response to the Comment 2 of Reviewer 2. Finally, thank you again for your precious time and continuous support of our work.

References

- 1 Fang, C. *et al.* Quantifying inactive lithium in lithium metal batteries. *Nature* **572**, 511–515 (2019).
- 2 He, X. *et al.* The passivity of lithium electrodes in liquid electrolytes for secondary batteries. *Nat. Rev. Mater.* **6**, 1036–1052 (2021).
- 3 Shan, H. *et al.* Investigation of self-discharge properties and a new concept of open-circuit voltage drop rate in lithium-ion batteries. *J. Solid State Electrochem.* **26**, 163–170 (2021).
- 4 Luo, L. *et al.* Surface coating constraint induced self-discharging of silicon nanoparticles as anodes for lithium ion batteries. *Nano Lett.* **15**, 7016–7022 (2015).
- 5 Zilberman, I., Sturm, J. & Jossen, A. Reversible self-discharge and calendar aging of 18650 nickel-rich, silicon-graphite lithium-ion cells. *J. Power Sources* **425**, 217–226 (2019).
- 6 Yazami, R. & Reynier, Y. F. Mechanism of self-discharge in graphite–lithium anode. *Electrochim. Acta* **47**, 1217–1223 (2002).
- 7 Shi, P. *et al.* Inhibiting intercrystalline reactions of anode with electrolytes for long-cycling lithium batteries. *Sci. Adv.* **8**, eabq3445 (2022).
- 8 Gabryelczyk, A., Ivanov, S., Bund, A. & Lota, G. Corrosion of aluminium current collector in lithium-ion batteries: A review. *J. Energy Storage* **43**, 103226 (2021).
- 9 Peled, E. Film forming reaction at the lithium/electrolyte interface. *J. Power Sources* **9**, 253–266 (1983).
- 10 Peled, E. The electrochemical behavior of alkali and alkaline earth metals in nonaqueous battery systems—the solid electrolyte interphase model. *J. Electrochem. Soc.* **126**, 2047 (1979).
- 11 Eswarappa Prameela, S. *et al.* Materials for extreme environments. *Nat. Rev. Mater.* **8**, 81–88 (2022).
- 12 Zhang, X. *et al.* Uniform corrosion behavior of GZ51K alloy with long period stacking ordered structure for biomedical application. *Corros. Sci.* **88**, 1–5 (2014).
- 13 Frankel, G. S. Pitting corrosion of metals: A review of the critical factors. *J. Electrochem. Soc.* **145**, 2186 (1998).

- 14 Guo, X. *et al.* Self-accelerated corrosion of nuclear waste forms at material interfaces. *Nat. Mater.* **19**, 310–316 (2020).
- 15 Brau, F., Thouvenel-Romans, S., Steinbock, O., Cardoso, S. S. S. & Cartwright, J. H. E. Filiform corrosion as a pressure-driven delamination process. *Soft Matter* **15**, 803–812 (2019).
- 16 Clark, B., Clair, J. S. & Edwards, M. Copper deposition corrosion elevates lead release to potable water. *Journal AWWA* **107**, E627–E637 (2015).
- 17 Yan, C. *et al.* Evading strength-corrosion tradeoff in Mg alloys via dense ultrafine twins. *Nat. Commun.* **12**, 4616 (2021).
- 18 Michalske, T. A. & Freiman, S. W. A molecular interpretation of stress corrosion in silica *Nature* **295**, 511–512 (1982).
- 19 Olugbade, T. O., Omiyale, B. O. & Ojo, O. T. Corrosion, corrosion fatigue, and protection of magnesium alloys: Mechanisms, measurements, and mitigation. *J. Mater. eng. perform.* **31**, 1707–1727 (2021).
- 20 Song, J. & Curtin, W. A. Atomic mechanism and prediction of hydrogen embrittlement in iron. *Nat. Mater.* **12**, 145–151 (2013).
- 21 Wu, L., Ma, A., Zhang, L. & Zheng, Y. Intergranular erosion corrosion of pure copper tube in flowing NaCl solution. *Corros. Sci.* **201**, 110304 (2022).
- 22 Qin, Z. *et al.* Microstructure design to improve the corrosion and cavitation corrosion resistance of a nickel-aluminum bronze. *Corros. Sci.* **139**, 255–266 (2018).
- 23 Zahran, R. R. & Sedahmed, G. H. Mass-transfer-controlled impingement corrosion at the jet inlet zone of an annulus under turbulent flow. *Ind. Eng. Chem. Res* **45**, 1160–1166 (2006).
- 24 Kaufman, J. *et al.* The effect of laser shock peening with and without protective coating on intergranular corrosion of sensitized AA5083. *Corros. Sci.* **194**, 109925 (2022).
- 25 Schroer, C., Wedemeyer, O., Novotny, J., Skrypnik, A. & Konys, J. Selective leaching of nickel and chromium from Type 316 austenitic steel in oxygen-containing lead–bismuth eutectic (LBE). *Corros. Sci.* **84**, 113–124 (2014).
- 26 Guo, X. *et al.* Understanding the fretting corrosion mechanism of zirconium alloy exposed to high temperature high pressure water. *Corros. Sci.* **202**, 110300 (2022).

- 27 Nimmervoll, M., Mori, G., Hönig, S. & Haubner, R. High-temperature corrosion of austenitic alloys in HCl and H₂S containing atmospheres under reducing conditions. *Corros. Sci.* **200**, 110214 (2022).
- 28 Videla, H. e. A. Prevention and control of biocorrosion. *Int. Biodeterior. Biodegradation* **39**, 259–270 (2002).
- 29 Yang, Y. *et al.* One dimensional wormhole corrosion in metals. *Nat. Commun.* **14**, 988 (2023).
- 30 Zhao, H. *et al.* Hydrogen trapping and embrittlement in high-strength Al alloys. *Nature* **602**, 437–441 (2022).
- 31 Moore, S. *et al.* Observation of stress corrosion cracking using real-time in situ high-speed atomic force microscopy and correlative techniques. *NPJ Mater. Degrad.* **5**, 3 (2021).
- 32 Lin, A. Y. W., Muller, A., Yu, X. X., Minor, A. M. & Marks, L. D. Early-stage NiCrMo oxidation revealed by cryo-transmission electron microscopy. *Ultramicroscopy* **200**, 6–11 (2019).
- 33 Boyle, D. T. *et al.* Corrosion of lithium metal anodes during calendar ageing and its microscopic origins. *Nat. Energy* **6**, 487–494 (2021).
- 34 Mouanga, M. *et al.* Galvanic corrosion between zinc and carbon steel investigated by local electrochemical impedance spectroscopy. *Electrochim. Acta* **88**, 6–14 (2013).
- 35 Manhabosco, S. M. *et al.* Localized corrosion of laser marked M340 martensitic stainless steel for biomedical applications studied by the scanning vibrating electrode technique under polarization. *Electrochim. Acta* **200**, 189–196 (2016).
- 36 Miller, R. B. *et al.* Novel mechanism of microbially induced carbon steel corrosion at an aqueous/non-aqueous interface. *Ind. Eng. Chem. Res.* **59**, 15784–15790 (2020).
- 37 Dong, C. *et al.* Integrated computation of corrosion: Modelling, simulation and applications. *Corros. Commun.* **2**, 8–23 (2021).
- 38 Wei, X. X. *et al.* Enhanced corrosion resistance by engineering crystallography on metals. *Nat. Commun.* **13**, 726 (2022).
- 39 Wang, Y. M. *et al.* Additively manufactured hierarchical stainless steels with high strength and ductility. *Nat. Mater.* **17**, 63–71 (2018).
- 40 Zhang, S. *et al.* Concerning the stability of seawater electrolysis: a corrosion

- mechanism study of halide on Ni-based anode. *Nat. Commun.* **14**, 4822 (2023).
- 41 Luo, H. *et al.* A strong and ductile medium-entropy alloy resists hydrogen embrittlement and corrosion. *Nat. Commun.* **11**, 3081 (2020).
- 42 Bastos, A. C., Quevedo, M. C., Karavai, O. V. & Ferreira, M. G. S. Review—on the application of the scanning vibrating electrode technique (SVET) to corrosion research. *J. Electrochem. Soc.* **164**, C973–C990 (2017).
- 43 Collins, L. *et al.* Probing charge screening dynamics and electrochemical processes at the solid-liquid interface with electrochemical force microscopy. *Nat. Commun.* **5**, 3871 (2014).
- 44 Jadhav, N. & Gelling, V. J. Review—the use of localized electrochemical techniques for corrosion studies. *J. Electrochem. Soc.* **166**, C3461–C3476 (2019).
- 45 Polcari, D., Dauphin-Ducharme, P. & Mauzeroll, J. Scanning electrochemical microscopy: A comprehensive review of experimental parameters from 1989 to 2015. *Chem. Rev.* **116**, 13234–13278 (2016).
- 46 Xu, W. *et al.* A high-specific-strength and corrosion-resistant magnesium alloy. *Nat. Mater.* **14**, 1229–1235 (2015).
- 47 Yang, L. & Yang, A. A. Communication—on zero-resistance ammeter and zero-voltage ammeter. *J. Electrochem. Soc.* **13**, C819–C821 (2017).
- 48 Raja, V. S. Grand challenges in metal corrosion and protection research. *Front. Met. Alloy.* **1**, 894181 (2022).
- 49 Leung, K., Merrill, L. C. & Harrison, K. L. Galvanic corrosion and electric field in lithium anode passivation films: Effects on self-discharge. *J. Phys. Chem. C* **126**, 8565–8580 (2022).
- 50 Lin, D. *et al.* Fast galvanic lithium corrosion involving a Kirkendall-type mechanism. *Nat. Chem.* **11**, 382–389 (2019).
- 51 Ding, J. F. *et al.* Dynamic galvanic corrosion of working lithium metal anode under practical conditions. *Adv. Energy Mater.* **13**, 2204305 (2023).
- 52 Gao, H. *et al.* Visualizing the failure of solid electrolyte under GPa-level interface stress induced by lithium eruption. *Nat. Commun.* **13**, 5050 (2022).
- 53 Yang, S. *et al.* Corrosion inhibition of aluminum current collector with molybdate conversion coating in commercial LiPF₆-esters electrolytes. *Corros. Sci.* **190**, 109632 (2021).

REVIEWERS' COMMENTS

Reviewer #2 (Remarks to the Author):

The authors have exhaustively responded to my questions. I have no further comments and recommend publication in Nature Communications.